# Bridging Mode Connectivity in Loss Landscapes and Adversarial Robustness

**Pu Zhao[1], Pin-Yu Chen[2], Payel Das[2], Karthikeyan Natesan Ramamurthy[2], Xue Lin[1]**
[1]Northeastern University, Boston, MA 02115
[2]IBM Research, Yorktown Heights, NY 10598
zhao.pu@husky.neu.edu, pin-yu.chen@ibm.com, daspa@us.ibm.com,
knatesa@us.ibm.com, xue.lin@northeastern.edu

## Abstract

Mode connectivity provides novel geometric insights on analyzing loss landscapes and enables building high-accuracy pathways between well-trained neural networks. In this work, we propose to employ mode connectivity in loss landscapes to study the adversarial robustness of deep neural networks, and provide novel methods for improving this robustness. Our experiments cover various types of adversarial attacks applied to different network architectures and datasets. When network models are tampered with backdoor or error-injection attacks, our results demonstrate that the path connection learned using limited amount of bonafide data can effectively mitigate adversarial effects while maintaining the original accuracy on clean data. Therefore, mode connectivity provides users with the power to repair backdoored or error-injected models. We also use mode connectivity to investigate the loss landscapes of regular and robust models against evasion attacks. Experiments show that there exists a barrier in adversarial robustness loss on the path connecting regular and adversarially-trained models. A high correlation is observed between the adversarial robustness loss and the largest eigenvalue of the input Hessian matrix, for which theoretical justifications are provided. Our results suggest that mode connectivity offers a holistic tool and practical means for evaluating and improving adversarial robustness[1].

## 1 Introduction

Recent studies on mode connectivity show that two independently trained deep neural network (DNN) models with the same architecture and loss function can be connected on their loss landscape using a high-accuracy/low-loss path characterized by a simple curve (Garipov et al., 2018; Gotmare et al., 2018; Draxler et al., 2018). This insight on the loss landscape geometry provides us with easy access to a large number of similar-performing models on the low-loss path between two given models, and Garipov et al. (2018) use this to devise a new model ensembling method. Another line of recent research reveals interesting geometric properties relating to adversarial robustness of DNNs (Fawzi et al., 2017; 2018; Wang et al., 2018b; Yu et al., 2018). An adversarial data or model is defined to be one that is close to a bonafide data or model in some space, but exhibits unwanted or malicious behavior. Motivated by these geometric perspectives, in this study, we propose to employ mode connectivity to study and improve adversarial robustness of DNNs against different types of threats.

A DNN can be possibly tampered by an adversary during different phases in its life cycle. For example, during the training phase, the training data can be corrupted with a designated trigger pattern associated with a target label to implant a backdoor for trojan attack on DNNs (Gu et al., 2019; Liu et al., 2018). During the inference phase when a trained model is deployed for task-solving, prediction-evasive attacks are plausible (Biggio & Roli, 2018; Goodfellow et al., 2015; Zhao et al., 2018), even when the model internal details are unknown to an attacker (Chen et al., 2017; Ilyas et al., 2018; Zhao et al., 2019a). In this research, we will demonstrate that by using mode connectivity in loss landscapes, we can repair backdoored or error-injected DNNs. We also show that mode

---

[1]The code is available at `https://github.com/IBM/model-sanitization`

connectivity analysis reveals the existence of a robustness loss barrier on the path connecting regular and adversarially-trained models.

We motivate the novelty and benefit of using mode connectivity for mitigating training-phase adversarial threats through the following practical scenario: as training DNNs is both time- and resource-consuming, it has become a common trend for users to leverage pre-trained models released in the public domain[2]. Users may then perform model fine-tuning or transfer learning with a small set of bonafide data that they have. However, publicly available pre-trained models may carry an unknown but significant risk of tampering by an adversary. It can also be challenging to detect this tampering, as in the case of a backdoor attack[3], since a backdoored model will behave like a regular model in the absence of the embedded trigger. Therefore, it is practically helpful to provide tools to users who wish to utilize pre-trained models while mitigating such adversarial threats. We show that our proposed method using mode connectivity with limited amount of bonafide data can repair backdoored or error-injected DNNs, while greatly countering their adversarial effects.

Our main contributions are summarized as follows:

- For backdoor and error-injection attacks, we show that the path trained using limited bonafide data connecting two tampered models can be used to repair and redeem the attacked models, thereby resulting in high-accuracy and low-risk models. The performance of mode connectivity is significantly better than several baselines including fine-tuning, training from scratch, pruning, and random weight perturbations. We also provide technical explanations for the effectiveness of our path connection method based on model weight space exploration and similarity analysis of input gradients for clean and tampered data.

- For evasion attacks, we use mode connectivity to study standard and adversarial-robustness loss landscapes. We find that between a regular and an adversarially-trained model, training a path with standard loss reveals no barrier, whereas the robustness loss on the same path reveals a barrier. This insight provides a geometric interpretation of the "no free lunch" hypothesis in adversarial robustness (Tsipras et al., 2019; Dohmatob, 2018; Bubeck et al., 2019). We also provide technical explanations for the high correlation observed between the robustness loss and the largest eigenvalue of the input Hessian matrix on the path.

- Our experimental results on different DNN architectures (ResNet and VGG) and datasets (CIFAR-10 and SVHN) corroborate the effectiveness of using mode connectivity in loss landscapes to understand and improve adversarial robustness. We also show that our path connection is resilient to the considered adaptive attacks that are aware of our defense. To the best of our knowledge, this is the first work that proposes using mode connectivity approaches for adversarial robustness.

## 2 BACKGROUND AND RELATED WORK

### 2.1 MODE CONNECTIVITY IN LOSS LANDSCAPES

Let $w_1$ and $w_2$ be two sets of model weights corresponding to two neural networks independently trained by minimizing any user-specified loss $l(w)$, such as the cross-entropy loss. Moreover, let $\phi_\theta(t)$ with $t \in [0, 1]$ be a continuous piece-wise smooth parametric curve, with parameters $\theta$, such that its two ends are $\phi_\theta(0) = w_1$ and $\phi_\theta(1) = w_2$.

To find a high-accuracy path between $w_1$ and $w_2$, it is proposed to find the parameters $\theta$ that minimize the expectation over a uniform distribution on the curve (Garipov et al., 2018),

$$L(\theta) = E_{t \sim q_\theta(t)} \left[ l(\phi_\theta(t)) \right] \tag{1}$$

where $q_\theta(t)$ is the distribution for sampling the models on the path indexed by $t$.

Since $q_\theta(t)$ depends on $\theta$, in order to render the training of high-accuracy path connection more computationally tractable, (Garipov et al., 2018; Gotmare et al., 2018) proposed to instead use the following loss term,

$$L(\theta) = E_{t \sim U(0,1)} \left[ l(\phi_\theta(t)) \right] \tag{2}$$

---

[2]For example, the Model Zoo project: `https://modelzoo.co`

[3]See the recent call for proposals on Trojans in AI announced by IARPA: `https://www.iarpa.gov/index.php/research-programs/trojai/trojai-baa`

where $U(0, 1)$ is the uniform distribution on $[0, 1]$.

The following functions are commonly used for characterizing the parametric curve function $\phi_\theta(t)$. **Polygonal chain (Gomes et al., 2012).** The two trained networks $w_1$ and $w_2$ serve as the endpoints of the chain and the bends of the chain are parameterized by $\theta$. For instance, the case of a chain with one bend is

$$\phi_\theta(t) = \begin{cases} 2\left(t\theta + (0.5 - t)\,\omega_1\right), & 0 \leq t \leq 0.5 \\ 2\left((t - 0.5)\,\omega_2 + (1 - t)\,\theta\right), & 0.5 \leq t \leq 1. \end{cases} \tag{3}$$

**Bezier curve (Farouki, 2012).** A Bezier curve provides a convenient parametrization of smoothness on the paths connecting endpoints. For instance, a quadratic Bezier curve with endpoints $w_1$ and $w_2$ is given by

$$\phi_\theta(t) = (1 - t)^2 \omega_1 + 2t\,(1 - t)\,\theta + t^2 \omega_2, \quad 0 \leq t \leq 1. \tag{4}$$

It is worth noting that, while current research on mode connectivity mainly focuses on generalization analysis (Garipov et al., 2018; Gotmare et al., 2018; Draxler et al., 2018; Wang et al., 2018a) and has found remarkable applications such as fast model ensembling (Garipov et al., 2018), our results show that its implication on adversarial robustness through the lens of loss landscape analysis is a promising, yet largely unexplored, research direction. Yu et al. (2018) scratched the surface but focused on interpreting decision surface of input space and only considered evasion attacks.

## 2.2 BACKDOOR, EVASION, AND ERROR-INJECTION ADVERSARIAL ATTACKS

**Backdoor attack.** Backdoor attack on DNNs is often accomplished by designing a designated trigger pattern with a target label implanted to a subset of training data, which is a specific form of data poisoning (Biggio et al., 2012; Shafahi et al., 2018; Jagielski et al., 2018). A backdoored model trained on the corrupted data will output the target label for any data input with the trigger; and it will behave as a normal model when the trigger is absent. For mitigating backdoor attacks, majority of research focuses on backdoor detection or filtering anomalous data samples from training data for re-training (Chen et al., 2018; Wang et al., 2019; Tran et al., 2018), while our aim is to repair backdoored models for models using mode connectivity and limited amount of bonafide data.

**Evasion attack.** Evasion attack is a type of inference-phase adversarial threat that generates adversarial examples by mounting slight modification on a benign data sample to manipulate model prediction (Biggio & Roli, 2018). For image classification models, evasion attack can be accomplished by adding imperceptible noises to natural images and resulting in misclassification (Goodfellow et al., 2015; Carlini & Wagner, 2017; Xu et al., 2018). Different from training-phase attacks, evasion attack does not assume access to training data. Moreover, it can be executed even when the model details are unknown to an adversary, via black-box or transfer attacks (Papernot et al., 2017; Chen et al., 2017; Zhao et al., 2020).

**Error-injection attack.** Different from attacks modifying data inputs, error-injection attack injects errors to model weights at the inference phase and aims to cause misclassification of certain input samples (Liu et al., 2017; Zhao et al., 2019b). At the hardware level of a deployed machine learning system, it can be made plausible via laser beam (Barenghi et al., 2012) and row hammer (Van Der Veen et al., 2016) to change or flip the logic values of the corresponding bits and thus modifying the model parameters saved in memory.

## 3 MAIN RESULTS

Here we report the experimental results, provide technical explanations, and elucidate the effectiveness of using mode connectivity for studying and enhancing adversarial robustness in three representative themes: (i) backdoor attack; (ii) error-injection attack; and (iii) evasion attack. Our experiments were conducted on different network architectures (VGG and ResNet) and datasets (CIFAR-10 and SVHN). The details on experiment setups are given in Appendix A. When connecting models, we use the cross entropy loss and the quadratic Bezier curve as described in (4). In what follows, we begin by illustrating the problem setups bridging mode connectivity and adversarial robustness, summarizing the results of high-accuracy (low-loss) pathways between untampered models for reference, and then delving into detailed discussions. Depending on the context, we will use the terms error rate

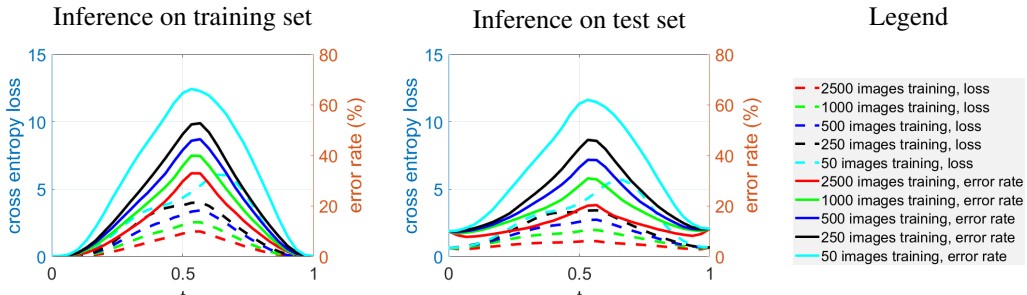

Figure 1: Loss and error rate on the path connecting two untampered VGG models trained on CIFAR-10. The path connection is trained using different settings as indicated by the curve colors. The results using SVHN and ResNet are given in Appendix B. The inference results on test set are evaluated using 5000 samples, which are separate from what are used for path connection.

Table 1: Error rate of backdoored models. The error rate of clean/backdoored samples means standard-test-error/attack-failure-rate, respectively. The results are evaluated on 5000 non-overlapping clean/triggered images selected from the test set. For reference, the test errors of clean images on untampered models are 12% for CIFAR-10 (VGG), and 4% for SVHN (ResNet), respectively.

| | Backdoor attacks | Single-target attack | | All-targets attack | |
| | Dataset | CIFAR-10 (VGG) | SVHN (ResNet) | CIFAR-10 (VGG) | SVHN (ResNet) |
|---|---|---|---|---|---|
| Model ($t = 0$) | Clean images | 15% | 5.4% | 14.2% | 6.1% |
| | Triggered images | 0.07% | 0.22% | 12.9% | 8.3% |
| Model ($t = 1$) | Clean images | 13% | 7.7% | 19% | 7.5% |
| | Triggered images | 2% | 0.17% | 13.6% | 9.2% |

and accuracy on clean/adversarial samples interchangeably. The error rate of adversarial samples is equivalent to their attack failure rate as well as 100%- attack accuracy.

## 3.1 Problem Setup and Mode Connection between Untampered Models

**Problem setup for backdoor and error-injection attacks.** We consider the practical scenario as motivated in Section 1, where a user has two potentially tampered models and a limited number of bonafide data at hand. The tampered models behave normally as untampered ones on non-triggered/non-targeted inputs so the user aims to fully exploit the model power while alleviating adversarial effects. The problem setup applies to the case of one tampered model, where we use the bonafide data to train a fine-tuned model and then connect the given and the fine-tuned models.

**Problem setup for evasion attack.** For gaining deeper understanding on evasion attacks, we consider the scenario where a user has access to the entire training dataset and aims to study the behavior of the models on the path connecting two independently trained models in terms of standard and robust loss landscapes, including model pairs selected from regular and adversarially-trained models.

**Regular path connection between untampered models.** Figure 1 shows the cross entropy loss and training/test error rate of models on the path connecting untampered models. The untampered models are independently trained using the entire training data. While prior results have demonstrated high-accuracy path connection using the entire training data (Garipov et al., 2018; Gotmare et al., 2018; Draxler et al., 2018), our path connection is trained using different portion of the original test data corresponding to the scenario of limited amount of bonafide data. Notably, when connecting two DNNs, a small number of clean data is capable of finding models with good performance. For example, path connection using merely 1000/2500 CIFAR-10 samples only reduces the test accuracy (on other 5000 samples) of VGG16 models by at most 10%/5% when compared to the well-trained models (those at $t = 0$ and $t = 1$), respectively. In addition, regardless of the data size used for path connection, the model having the worst performance is usually located around the point $t = 0.5$, as it is geometrically the farthest model from the two end models on the path.

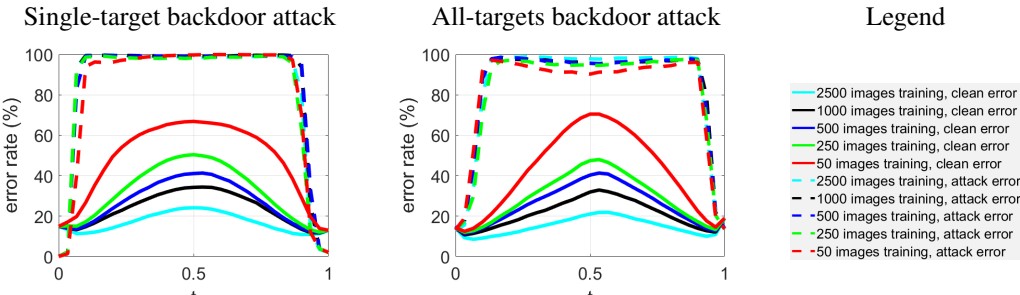

Figure 2: Error rate against backdoor attacks on the connection path for CIFAR-10 (VGG). The error rate of clean/backdoored samples means the standard-test-error/attack-failure-rate, respectively.

## 3.2 MITIGATING AND REPAIRING BACKDOORED MODELS

**Attack implementation.** We follow the procedures in (Gu et al., 2019) to implement backdoor attacks and obtain two backdoored models trained on the same poisoned training data. The trigger pattern is placed at the right-bottom of the poisoned images as shown in Appendix C. Specifically, 10% of the training data are poisoned by inserting the trigger and changing the original correct labels to the target label(s). Here we investigate two kinds of backdoor attacks: (a) single-target attack which sets the target label $T$ to a specific label (we choose $T$ = class 1); and (b) all-targets attack where the target label $T$ is set to the original label $i$ plus 1 and then modulo 9, i.e., $T = i + 1 \pmod 9$. Their performance on clean (untriggered) and triggered data samples are given in Table 1, and the prediction errors of triggered images relative to the true labels are given in Appendix D. The backdoored models have similar performance on clean data as untampered models but will indeed misclassify majority of triggered samples. Comparing to single-target attack, all-targets attack is more difficult and has a higher attack failure rate, since the target labels vary with the original labels.

**Evaluation and analysis.** We train a path connecting the two backdoored models with limited amount of bonafide data. As shown in Figure 2, at both path endpoints ($t = \{0, 1\}$) the two tampered models attain low error rates on clean data but are also extremely vulnerable to backdoor attacks (low error rate on backdoored samples means high attack success rate). Nonetheless, we find that path connection with limited bonafide data can effectively mitigate backdoor attacks and redeem model power. For instance, the models at $t = 0.1$ or $t = 0.9$ can simultaneously attain similar performance on clean data as the tampered models while greatly reducing the backdoor attack success rate from close to 100% to nearly 0%. Moreover, most models on the path (e.g. when $t \in [0.1, 0.9]$) exhibit high resilience to backdoor attacks, suggesting mode connection with limited amount of bonafide data can be an effective countermeasure. While having resilient models to backdoor attacks on the path, we also observe that the amount of bonafide data used for training path has a larger impact on the performance of clean data. Path connection using fewer data samples will yield models with higher error rates on clean data, which is similar to the results of path connection between untampered models discussed in Section 3.1. The advantages of redeeming model power using mode connectivity are consistent when evaluated on different network architectures and datasets (see Appendix E).

**Comparison with baselines.** We compare the performance of mode connectivity against backdoor attacks with the following baseline methods: (i) fine-tuning backdoored models with bonafide data; (ii) training a new model of the same architecture from scratch with bonafide data; (iii) model weight pruning and then fine-tuning with bonafide data using (Li et al., 2017); and (iv) random Gaussian perturbation to the model weights leading to a noisy model. The results are summarized in Table 2 and their implementation details are given in Appendix E. Evaluated on different network architectures and datasets, the path connection method consistently maintains superior accuracy on clean data while simultaneously attaining low attack accuracy over the baseline methods, which can be explained by the ability of finding high-accuracy paths between two models using mode connectivity. For CIFAR-10 (VGG), even using as few as 50 bonafide samples for path connection, the subsequent model in Table 2 still remains 63% clean accuracy while constraining backdoor accuracy to merely 2.5%. The best baseline method is fine-tuning, which has similar backdoor accuracy as path connection but attains lower clean accuracy (e.g. 17% worse than path connection when using 50 bonafide samples). For SVHN (ResNet), the clean accuracy of fine-tuning can be on par with path connection, but its backdoor accuracy is significantly higher than path connection. For example, when trained with 250

Table 2: Performance against single-target backdoor attack. The clean/backdoor accuracy means standard-test-accuracy/attack-success-rate, respectively. More results are given in Appendix E.

| | | Method / Bonafide data size | 2500 | 1000 | 500 | 250 | 50 |
|---|---|---|---|---|---|---|---|
| CIFAR-10 (VGG) | Clean Accuracy | Path connection ($t = 0.1$) | 88% | 83% | 80% | 77% | 63% |
| | | Fine-tune | 84% | 82% | 78% | 74% | 46% |
| | | Train from scratch | 50% | 39% | 31% | 30% | 20% |
| | | Noisy model ($t = 0$) | 21% | 21% | 21% | 21% | 21% |
| | | Noisy model ($t = 1$) | 24% | 24% | 24% | 24% | 24% |
| | | Prune | 88% | 85% | 83% | 82% | 81% |
| | Backdoor Accuracy | Path connection ($t = 0.1$) | 1.1% | 0.8% | 1.5% | 3.3% | 2.5% |
| | | Fine-tune | 1.5% | 0.9% | 0.5% | 1.9% | 2.8% |
| | | Train from scratch | 0.4% | 0.7% | 0.3% | 3.2% | 2.1% |
| | | Noisy model ($t = 0$) | 97% | 97% | 97% | 97% | 97% |
| | | Noisy model ($t = 1$) | 91% | 91% | 91% | 91% | 91% |
| | | Prune | 43% | 49% | 81% | 79% | 82% |
| SVHN (ResNet) | Clean Accuracy | Path connection ($t = 0.2$) | 96% | 94% | 93% | 89% | 82% |
| | | Fine-tune | 96% | 94% | 91% | 89% | 76% |
| | | Train from scratch | 87% | 75% | 61% | 34% | 12% |
| | | Noisy model ($t = 0$) | 13% | 13% | 13% | 13% | 13% |
| | | Noisy model ($t = 1$) | 11% | 11% | 11% | 11% | 11% |
| | | Prune | 96% | 95% | 93% | 91% | 89% |
| | Backdoor Accuracy | Path connection ($t = 0.2$) | 2.5% | 3% | 3.6% | 4.3% | 16% |
| | | Fine-tune | 14% | 7% | 29% | 63% | 60% |
| | | Train from scratch | 3% | 3.6% | 5% | 2.2% | 3.9% |
| | | Noisy model ($t = 0$) | 51% | 51% | 51% | 51% | 51% |
| | | Noisy model ($t = 1$) | 42% | 42% | 42% | 42% | 42% |
| | | Prune | 80% | 90% | 88% | 92% | 94% |

samples, they have the same clean accuracy but the backdoor accuracy of fine-tuning is 58.7% higher than path connection. Training from scratch does not yield competitive results given limited amount of training data. Noisy models perturbed by adding zero-mean Gaussian noises to the two models are not effective against backdoor attacks and may suffer from low clean accuracy. Pruning gives high clean accuracy but has very little effect on mitigating backdoor accuracy.

**Extensions.** Our proposal of using mode connectivity to repair backdoor models can be extended to the case when only one tampered model is given. We propose to fine-tune the model using bonafide data and then connect the given model with the fine-tuned model. Similar to the aforementioned findings, path connection can remain good accuracy on clean data while becoming resilient to backdoor attacks. We refer readers to Appendix G for more details. In addition, we obtain similar conclusions when the two backdoored models are trained with different poisoned datasets.

**Technical Explanations.** To provide technical explanations for the effectiveness of our proposed path connection method in repairing backdoored models, we run two sets of analysis: (i) model weight space exploration and (ii) data similarity comparison. For (i), we generate 1000 noisy versions of a backdoored model via random Gaussian weight perturbations. We find that they suffer from low clean accuracy and high attack success rate, which suggests that a good model with high-clean-accuracy and low-attack-accuracy is unlikely to be found by chance. We also report the distinct difference between noisy models and models on the path in the weight space to validate the necessity of using our path connection for attack mitigation and model repairing. More details are given in Appendix H. For (ii), we run similarity analysis of the input gradients between the end (backdoored) models and models on the connection path for both clean data and triggered data. We find that the similarity of triggered data is much lower than that of clean data when the model is further away in the path from the end models, suggesting that our path connection method can neutralize the backdoor effect. More details are given in Appendix I. The advantage of our path connection method over fine-tuning demonstrates the importance of using the knowledge of mode connectivity for model repairing.

**Adaptive Attack.** To justify the robustness of our proposed path connection approach to adaptive attacks, we consider the advanced attack setting where the attacker knows path connection is used for defense but cannot compromise the bonafide data that are private to an user. Furthermore, we allow

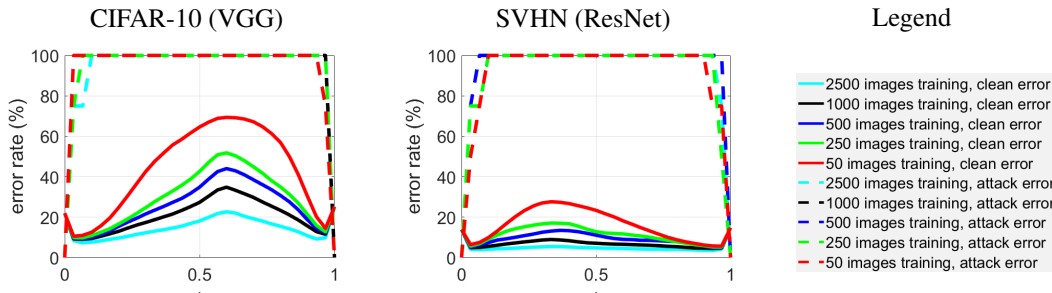

Figure 3: Error rate against error-injection attack on the connection path for CIFAR-10 (VGG). The error rate of clean/targeted samples means standard-test-error/attack-failure-rate, respectively.

Table 3: Performance against error-injection attack. The clean/injection accuracy means standard-test-accuracy/attack-success-rate, respectively. Path connection has the best clean accuracy and can completely remove injected errors (i.e. 0% attack accuracy). More results are given in Appendix F.

| Dataset | Method / Bonafide data size | Clean Accuracy | | | | | Injection Accuracy | | | | |
|---|---|---|---|---|---|---|---|---|---|---|---|
| | | 2500 | 1000 | 500 | 250 | 50 | 2500 | 1000 | 500 | 250 | 50 |
| | Path connection ($t = 0.1$) | 92% | 90% | 90% | 90% | 88% | 0% | 0% | 0% | 0% | 0% |
| | Fine-tune | 88% | 87% | 86% | 84% | 82% | 0% | 0% | 0% | 0% | 0% |
| CIFAR-10 | Train from scratch | 45% | 37% | 27% | 25% | 10% | 0% | 0% | 0% | 0% | 0% |
| (VGG) | Noisy model ($t = 0$) | 14% | 14% | 14% | 14% | 14% | 36% | 36% | 36% | 36% | 36% |
| | Noisy model ($t = 1$) | 12% | 12% | 12% | 12% | 12% | 19% | 19% | 19% | 19% | 19% |
| | Prune | 91% | 89% | 88% | 88% | 88% | 0% | 0% | 25% | 25% | 25% |
| | Path connection ($t = 0.1$) | 96% | 94% | 92% | 91% | 90% | 0% | 0% | 0% | 0% | 0% |
| | Fine-tune | 94% | 93% | 91% | 89% | 88% | 0% | 25% | 0% | 25% | 25% |
| SVHN | Train from scratch | 90% | 83% | 75% | 61% | 21% | 0% | 0% | 0% | 0% | 0% |
| (ResNet) | Noisy model ($t = 0$) | 11% | 11% | 11% | 11% | 11% | 28% | 28% | 28% | 28% | 28% |
| | Noisy model ($t = 1$) | 11% | 11% | 11% | 11% | 11% | 18% | 18% | 18% | 18% | 18% |
| | Prune | 95% | 93% | 92% | 90% | 89% | 0% | 0% | 25% | 0% | 25% |

the attacker to use the *same* path training loss function as the defender. To attempt breaking path connection, the attacker trains a compromised path such that every model on this path is a backdoored model and then releases the path-aware tampered models. We show that our approach is still resilient to this adaptive attack. More details are given in Appendix J.

### 3.3 SANITIZING ERROR-INJECTED MODELS

**Attack implementation.** We adopt the fault sneaking attack (Zhao et al., 2019b) for injecting errors to model weights. Given two untampered and independently trained models, the errors are injected with selected samples as targets such that the tampered models will cause misclassification on targeted inputs and otherwise will behave as untampered models. More details are given in Appendix C.

**Evaluation and analysis.** Similar to the setup in Section 3.2, Figure 3 shows the clean accuracy and attack accuracy of the models on the path connecting two error-injected models using limited amount of bonafide data. For the error-injected models ($t = \{0, 1\}$), the attack accuracy is nearly 100%, which corresponds to 0% attack failure rate on targeted samples. However, using path connection and limited amount of bonafide data, the injected errors can be removed almost completely. Varying the size of path training data consistently sanitizes the error-injected models and mainly affects the standard test error. Most of the models on the path can attain nearly 100% fault tolerance (i.e. 100% attack failure rate) to the injected errors. The models on the path near $t = 0$ or $t = 1$ have comparable performance on clean data and exhibit strong fault tolerance to injected errors. Similar findings are observed across different network architectures and datasets (see Appendix F).

**Comparison with baselines and extensions.** In Table 3, we adopt the same baselines as in Section 3.2 to compare with path connection. We find that only path connection and training-from-scratch can successfully sanitize the error-injected models and attain 0% attack accuracy, and other baselines are less effective. Table 3 also shows the clean accuracy of path connection is substantially better than the effective baselines, suggesting novel applications of mode connectivity for finding accurate and adversarially robust models. The extensions to other settings are discussed in Appendix G.

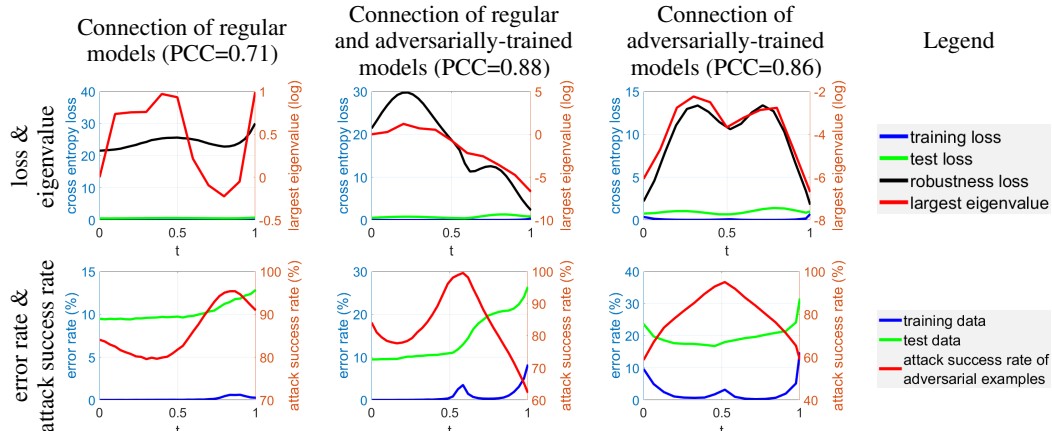

Figure 4: Loss, error rate, attack success rate and largest eigenvalue of input Hessian on the path connecting different model pairs on CIFAR-10 (VGG) using standard loss. The error rate of training/test data means standard training/test error, respectively. In all cases, there is no standard loss barrier but a robustness loss barrier. There is also a high correlation between the robustness loss and the largest eigenvalue of input Hessian, and their Pearson correlation coefficient (PCC) is reported in the title.

**Technical explanations and adaptive attack.** Consistent with the results in backdoor attacks, we explore the model weight space to demonstrate the significant difference between the models on our connection path and random noisy models. We also show that the similarity of error-injected images are much lower than that of clean images. In addition, our path connection is resilient to the advanced path-aware error-injection attack. More details are given in Appendices H, I and J.

### 3.4 Robustness Loss Landscape and Correlation Analysis for Evasion Attack

To gain insights on mode connectivity against evasion attacks, here we investigate the standard and adversarial-robustness loss landscapes on the same path connecting two untampered and independently trained models. The path is trained using the entire training dataset for minimizing equation 2 with standard cross entropy loss. The robustness loss refers to the cross entropy of class predictions on adversarial examples generated by evasion attacks and their original class labels. Higher robustness loss suggests the model is more vulnerable to evasion attacks. In addition, we will investigate the robustness loss landscape connecting regular (non-robust) and adversarially-trained (robust) models, where the path is also trained with standard cross entropy loss. We will also study the behavior of the largest eigenvalue of the Hessian matrix associated with the cross entropy loss and the data input, which we call the input Hessian. As adversarial examples are often generated by using the input gradients, we believe the largest eigenvalue of the input Hessian can offer new insights on robustness loss landscape, similar to the role of model-weight Hessian on quantifying generalization performance (Wu et al., 2017; Wang et al., 2018a).

**Attack Implementation.** We uniformly select 9 models ($t = \{0.1, 0.2, \ldots, 0.9\}$) on the path and run evasion attacks on each of them using the $\ell_\infty$-norm-ball based projected gradient descent (PGD) method proposed in (Madry et al., 2018). The robustness loss is evaluated using the non-targeted adversarial examples crafted from the entire test set, and the attack perturbation strength is set to $\epsilon = 8/255$ with 10 iterations. We also use the PGD method for adversarial training to obtain adversarially-trained models that are robust to evasion attacks but pay the price of reduced accuracy on clean data (Madry et al., 2018).

**Evaluation and Analysis.** To study the standard and robustness loss landscapes, we scrutinize the models on the path connecting the following pairs of models: (i) independently trained regular (non-robust) models; (ii) regular to adversarially-trained models; and (iii) independently adversarially-trained models. These results are shown in Figure 4. We summarize the major findings as follows.

- No standard loss barrier in all cases: Regardless of the model pairs, all models on the paths have similar standard loss metrics in terms of training and test losses, which are consistent with the previous results on the "flat" standard loss landscape for mode connectivity (Garipov et al., 2018; Gotmare et al., 2018; Draxler et al., 2018). The curve of standard loss in case (ii) is skewed toward

one end due to the artifact that the adversarially-trained model ($t = 1$) has a higher training/test error than the regular model ($t = 0$).

- **Robustness loss barrier**: Unlike standard loss, we find that the robustness loss on the connection path has a very distinct characteristic. In all cases, there is a robustness loss barrier (a hill) between pairs of regular and adversarially-trained models. The gap (height) of the robustness loss barrier is more apparent in cases (ii) and (iii). For (ii), the existence of a barrier suggests the modes of regular and adversarially-trained models are not connected by the path in terms of robustness loss, which also provides a geometrical evidence of the "no free lunch" hypothesis that adversarially robust models cannot be obtained without additional costs (Tsipras et al., 2019; Dohmatob, 2018; Bubeck et al., 2019). For (iii), robustness loss barriers also exist. The models on the path are less robust than the two adversarially-trained models at the path end, despite they have similar standard losses. The results suggest that there are essentially no better adversarially robust models on the path connected by regular training using standard loss.

- **High correlation between the largest eigenvalue of input Hessian and robustness loss**: Inspecting the largest eigenvalue of input Hessian $H_t(x)$ of a data input $x$ on the path, denoted by $\lambda_{\max}(t)$, we observe a strong accordance between $\lambda_{\max}(t)$ and robustness loss on the path, verified by the high empirical Pearson correlation coefficient (PCC) averaged over the entire test set as reported in Figure 4. As evasion attacks often use input gradients to craft adversarial perturbations to $x$, the eigenspectrum of input Hessian indicates its local loss curvature and relates to adversarial robustness (Yu et al., 2018). The details of computing $\lambda_{\max}(t)$ are given in Appendix K. Below we provide technical explanations for the empirically observed high correlation between $\lambda_{\max}(t)$ and the oracle robustness loss on the path, defined as $\max_{\|\delta\| \le \epsilon} l(w(t), x + \delta)$.

**Proposition 1.** *Let $f_w(\cdot)$ be a neural network classifier with its model weights denoted by $w$ and let $l(w, x)$ denote the classification loss (e.g. cross entropy of $f_w(x)$ and the true label $y$ of a data sample $x$). Consider the oracle robustness loss $\max_{\|\delta\| \le \epsilon} \ell(w(t), x + \delta)$ of the model $t$ on the path, where $\delta$ denotes a perturbation to $x$ confined by an $\epsilon$-ball induced by a vector norm $\| \cdot \|$. Assume (a) the standard loss $l(w(t), x)$ on the path is a constant for all $t \in [0, 1]$.*
*(b) $l(w(t), x + \delta) \approx l(w(t), x) + \nabla_x l(w(t), x)^T \delta + \frac{1}{2}\delta^T H_t(x)\delta$ for small $\delta$, where $\nabla_x l(w(t), x)$ is the input gradient and $H_t(x)$ is the input Hessian of $l(w(t), x)$ at $x$.*
*Let $c$ denote the normalized inner product in absolute value for the largest eigenvector $v$ of $H_t(x)$ and $\nabla_x l(w(t), x)$, $\frac{|\nabla_x l(w(t),x)^T v|}{\|\nabla_x l(w(t),x)\|} = c$. Then we have $\max_{\|\delta\| \le \epsilon} l(w(t), x + \delta) \sim \lambda_{\max}(t)$ as $c \to 1$.*

**Proof:** The proof is given in Appendix L. Assumption (a) follows by the existence of high-accuracy path of standard loss landscape from mode connectivity analysis. Assumption (b) assumes the local landscape with respect to the input $x$ can be well captured by its second-order curvature based on Taylor expansion. The value of $c$ is usually quite large, which has been empirically verified in both regular and adversarially-trained models (Moosavi-Dezfooli et al., 2019).

**Extensions.** Although we find that there is a robustness loss barrier on the path connected by regular training, we conduct additional experiments to show that it is possible to find an robust path connecting two adversarially-trained or regularly-trained model pairs using adversarial training (Madry et al., 2018), which we call the "robust connection" method. However, model ensembling using either the regular connection or robust connection has little gain against evasion attacks, as the adversarial examples are known to transfer between similar models (Papernot et al., 2016; Su et al., 2018). We refer readers to Appendix M for more details.

## 4 CONCLUSION

This paper provides novel insights on adversarial robustness of deep neural networks through the lens of mode connectivity in loss landscapes. Leveraging mode connectivity between model optima, we show that path connection trained by a limited number of clean data can successfully repair backdoored or error-injected models and significantly outperforms several baseline methods. Moreover, we use mode connectivity to uncover the existence of robustness loss barrier on the path trained by standard loss against evasion attacks. We also provide technical explanations for the effectiveness of our proposed approach and theoretically justify the empirically observed high correlation between robustness loss and the largest eigenvalue of input Hessian. Our findings are consistent and validated on different network architectures and datasets.

ACKNOWLEDGEMENTS

This work was primarily conducted during Pu Zhao's internship at IBM Research. This work is partly supported by the National Science Foundation CNS-1929300.

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

## APPENDIX

## A NETWORK ARCHITECTURE AND TRAINING

In this paper we mainly use two model architectures, VGG and ResNet. The VGG model (Simonyan & Zisserman, 2014) has 13 convolutional layers and 3 fully connected layers. The ResNet model is based on the Preactivation-ResNet implementation (He et al., 2016) with 26 layers. The clean test accuracy of untampered models of different architectures on CIFAR-10 and SVHN are given in Table A1. All of the experiments are performed on 6 GTX 1080Ti GPUs. The experiments are implemented with Python and Pytorch.

Table A1: Test accuracy of untampered models for different datasets and model architectures.

| | architecture | VGG | ResNet |
|---|---|---|---|
| CIFAR-10 | model ($t = 0$) | 88% | 87% |
| | model ($t = 1$) | 86% | 91% |
| SVHN | model ($t = 0$) | 96% | 97% |
| | model ($t = 1$) | 98% | 99% |

## B REGULAR PATH CONNECTION OF UNTAMPERED MODELS ON SVHN (RESNET)

The performance of regular path connection of untampered models on SVHN with ResNet is presented in Figure A1.

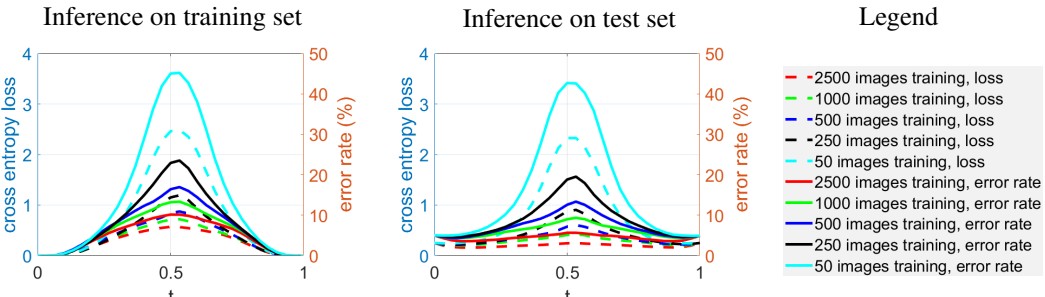

Figure A1: Loss and error rate on the path connecting two untampered ResNet models trained on SVHN. The path connection is trained using different settings as indicated by the curve colors. The inference results on test set are evaluated using 5000 samples, which are separate from what are used for path connection.

## C ILLUSTRATION AND IMPLEMENTATION DETAILS OF BACKDOOR AND ERROR-INJECTION ATTACKS

**Backdoor attack** The backdoor attack is implemented by poisoning the training dataset and then training a backdoored model with this training set. To poison the training set, we randomly pick 10% images from the training dataset and add a trigger to each image. The shape and location of the trigger is shown in Figure A2. Meanwhile, we set the labels of the triggered images to the target label(s) as described in Section 3.2.

**Error-injection attack** We select 1000 images from the test set and pick 4 images as targeted samples with randomly selected target labels for inducing attack. The target labels of the 4 selected images are different from their original correct labels. The goal of the attacker is to change the classification of the 4 images to the target labels while keeping the classification of the remaining 996 images unchanged through modifying the model parameters. To obtain the models with injected errors on CIFAR-10, we first train two models with a clean accuracy of 88% and 86%, respectively. Keeping

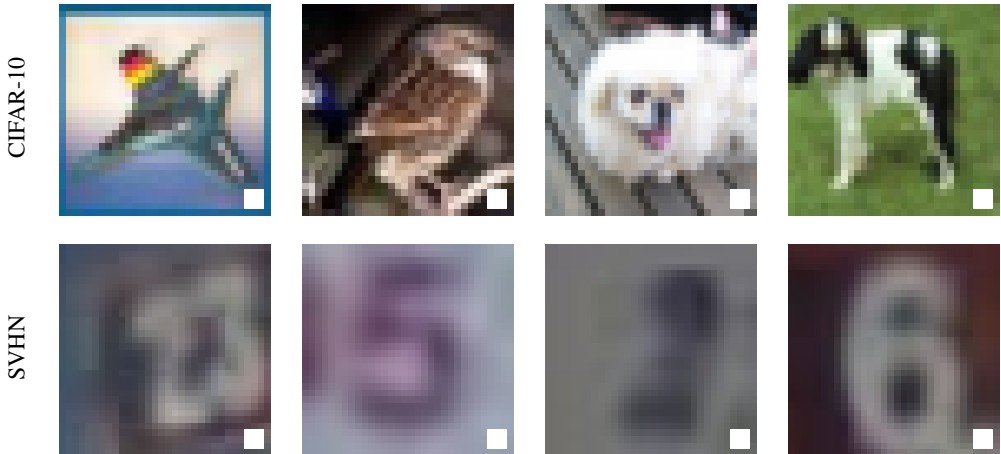

Figure A2: Examples of backdoored images on CIFAR-10 and SVHN. The triggers are white blocks located at the right-bottom area of each image.

the classification of a number of images unchanged can help to mitigate the accuracy degradation incurred by the model weight perturbation. After perturbing the model weights, the 4 errors can be injected into the model successfully with 100% accuracy for their target labels. The accuracy for other clean images become 78% and 75%, respectively.

## D   PREDICTION ERROR ON THE TRIGGERED DATA

We show the prediction error of the triggered data relative to the true labels on all datasets and networks in Figure A3. The error rate means the fraction of triggered images having top-1 predictions different from the original true labels. The prediction error rate of triggered data is high at path ends $(t = 0, 1)$ since the two end models are tampered. It has similar trend as standard test error for models not too close to the path ends, suggesting path connection can find models having good classification accuracy on triggered data.

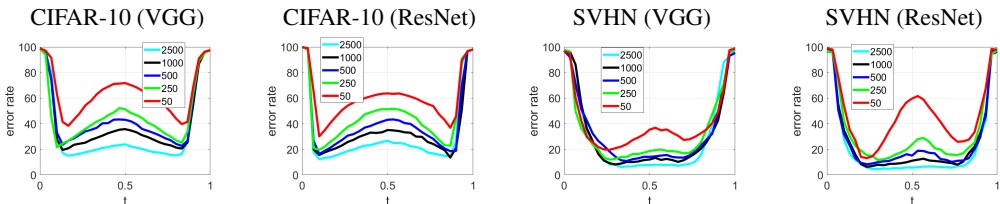

Figure A3: Prediction error rate against backdoor attacks on the connection path.

## E   MORE RESULTS ON PATH CONNECTION AGAINST BACKDOOR ATTACKS

**Implementation Details for Table 2**   For our proposed path connection method, we train the connection using different number of images as given in Table 2 for 100 epochs and then report the performance of the model associated with a selected index $t$ on the path. For the fine-tuning and training-from-scratch methods, we report the model performance after training for 100 epochs. For the random Gaussian perturbation to model weights, we evaluate the model performance under Gaussian noise perturbations on the model parameters. There are two given models which are the models at $t = 0$ and $t = 1$. The Gaussian noise has zero mean with a standard deviation of the absolute value of the difference between the two given models. Then we add the Gaussian noise to the two given models respectively and test their accuracy for clean and triggered images. For Gaussian noise, the experiment is performed multiple times (50 times) and we report the average accuracy. We can see that adding Gaussian noise perturbations to the model does not necessarily

change the model status from robust to non-robust or from non-robust to robust. The path connection or evolution from the model at $t = 0$ to that $t = 1$ follows a specific path achieving robustness against backdoor attack rather than random exploration. For pruning, we use the filter pruning method (Li et al., 2017) to prune filters from convolutional neural networks (CNNs) that are identified as having a small effect on the output accuracy. By removing the whole filters in the network together with their connecting feature maps, the computation costs are reduced significantly. We first prune about 60% of its parameters for VGG or 20% parameters for ResNet. Then we retrain the network with different number of images as given in Table 2 for 100 epochs. The clean accuracy and backdoor accuracy are as reported.

Figure A4 shows the error rates of clean and backdoored samples using CIFAR-10 on the connection path against single-target attacks.

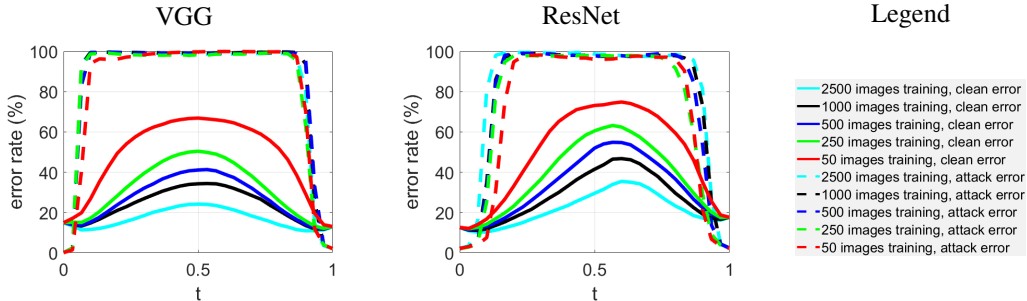

Figure A4: Error rate of single-target backdoor attack on the connection path for CIFAR-10. The error rate of clean/backdoored samples means standard-test-error/attack-failure-rate, respectively.

Figure A5 shows the error rates of clean and backdoored samples using SVHN on the connection path against single-target attacks.

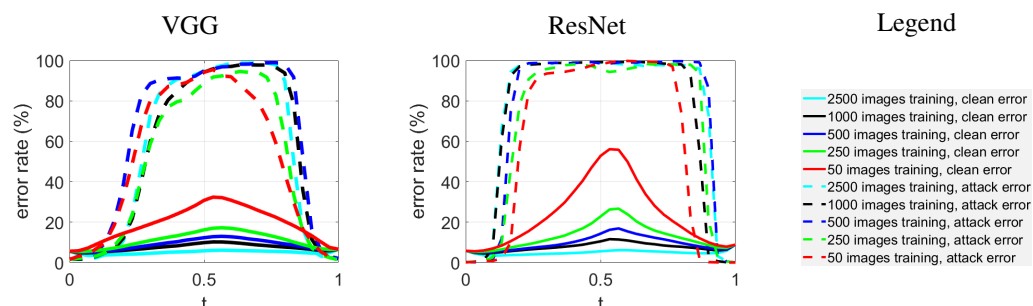

Figure A5: Error rate of single-target backdoor attack on the connection path for SVHN. The error rate of clean/backdoored samples means standard-test-error/attack-failure-rate, respectively.

Table A2 shows the performance comparison of path connection and other baseline methods against single-target backdoor attack evaluated on CIFAR-10 (ResNet) and SVHN (VGG).

## F    MORE RESULTS ON PATH CONNECTION AGAINST ERROR-INJECTION ATTACKS

**Implementation Details for Table 3**    For our proposed path connection method, we train the connection using different number of images as given in Table 3 for 100 epochs and then report the performance of the model associated with a selected index on the path. The start model and end model have been injected with the same 4 errors (misclassifying 4 given images), starting from two different unperturbed models obtained with different training hyper-parameters. For the fine-tuning and training-from-scratch methods, we report the model performance after training for 100 epochs. For the random Gaussian perturbation to model weights, we evaluate the model performance under Gaussian noise perturbations on the model parameters. There are two given models which are the

Table A2: Performance comparison of path connection and baselines against single-target backdoor attack. The clean/backdoor accuracy means standard-test-accuracy/attack-success-rate, respectively.

| | | Methods / bonafide data size | 2500 | 1000 | 500 | 250 | 50 |
|---|---|---|---|---|---|---|---|
| CIFAR-10 (ResNet) | Clean Accuracy | Path connection ($t = 0.2$) | 87% | 83% | 80% | 77% | 67% |
| | | Fine-tune | 85% | 84% | 83% | 83% | 72% |
| | | Train from scratch | 42% | 36% | 33% | 29% | 22% |
| | | Noisy model ($t = 0$) | 12% | 12% | 12% | 12% | 12% |
| | | Noisy model ($t = 1$) | 10% | 10% | 10% | 10% | 10% |
| | | Prune | 85% | 82% | 80% | 79% | 77% |
| | Backdoor Accuracy | Path connection ($t = 0.2$) | 0.6% | 1.2% | 1.3% | 2.3% | 5.3% |
| | | Fine-tune | 0.7% | 8.7% | 19% | 20% | 18% |
| | | Train from scratch | 3.7% | 4.3% | 3.5% | 8.2% | 6.5% |
| | | Noisy model ($t = 0$) | 95% | 95% | 95% | 95% | 95% |
| | | Noisy model ($t = 1$) | 97% | 97% | 97% | 97% | 97% |
| | | Prune | 37% | 52% | 78% | 86% | 88% |
| SVHN (VGG) | Clean Accuracy | Path connection ($t = 0.7$) | 95% | 93% | 91% | 87% | 75% |
| | | Fine-tune | 93% | 92% | 90% | 89% | 77% |
| | | Train from scratch | 88% | 82% | 76% | 54% | 25% |
| | | Noisy model ($t = 0$) | 22% | 22% | 22% | 22% | 22% |
| | | Noisy model ($t = 1$) | 16% | 16% | 16% | 16% | 16% |
| | | Prune | 94% | 92% | 91% | 89% | 86% |
| | Backdoor Accuracy | Path connection ($t = 0.7$) | 2% | 6% | 2% | 9% | 22% |
| | | Fine-tune | 11% | 10% | 13% | 30% | 61% |
| | | Train from scratch | 2.6% | 2.3% | 3.3% | 4.7% | 13% |
| | | Noisy model ($t = 0$) | 79% | 79% | 79% | 79% | 79% |
| | | Noisy model ($t = 1$) | 62% | 62% | 62% | 62% | 62% |
| | | Prune | 75% | 69% | 78% | 86% | 91% |

models at $t = 0$ and $t = 1$. The Gaussian noise has zero mean with a standard deviation of the absolute value of the difference between the two given models. Then we add the Gaussian noise to the two given models respectively and test their accuracy for clean and triggered images. The experiment is performed multiple times (50 times) and we report the average accuracy. We can see that adding Gaussian noise perturbations to the model does not necessarily change the model status from robust to non-robust or from non-robust to robust. The path connection or evolution from the model at $t = 0$ to that $t = 1$ follows a specific path achieving robustness against backdoor attack rather than random exploration. The training-from-scratch baseline method usually obtains the lowest clean test accuracy, especially in the case of training with 50 images, where training with so little images does not improve the accuracy.

Figure A6 shows the performance of path connection against error-injection attacks evaluated on CIFAR-10 (ResNet) and SVHN (VGG).

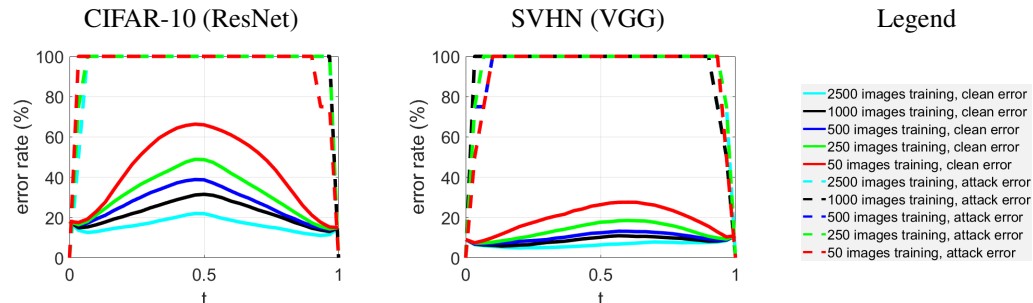

Figure A6: Error rate against error-injection attack on the connection path for CIFAR-10 (VGG). The error rate of clean/targeted samples means standard-test-error/attack-failure-rate, respectively.

Table 3 and A3 show the performance comparison of path connection and other baseline methods against error-injection attacks evaluated on all combinations of network architectures and datasets.

Table A3: Performance evaluation against error-injection attack. The clean/injection accuracy means standard-test-accuracy/attack-success-rate, respectively. Path connection attains the highest clean accuracy and lowest (0%) attack accuracy among all methods.

| | | Methods / bonafide data size | 2500 | 1000 | 500 | 250 | 50 |
|---|---|---|---|---|---|---|---|
| CIFAR-10 (ResNet) | Clean Accuracy | Path connection ($t = 0.1$) | 87% | 83% | 81% | 80% | 77% |
| | | Fine-tune | 80% | 79% | 78% | 75% | 72% |
| | | Train from scratch | 46% | 39% | 32% | 28% | 18% |
| | | Noisy model ($t = 0$) | 11% | 11% | 11% | 11% | 11% |
| | | Noisy model ($t = 1$) | 10% | 10% | 10% | 10% | 10% |
| | | Prune | 84% | 82% | 81% | 80% | 78% |
| | Injection Accuracy | Path connection ($t = 0.1$) | 0% | 0% | 0% | 0% | 0% |
| | | Fine-tune | 0% | 0% | 0% | 25% | 25% |
| | | Train from scratch | 0% | 0% | 0% | 0% | 0% |
| | | Noisy model ($t = 0$) | 33% | 33% | 33% | 33% | 33% |
| | | Noisy model ($t = 1$) | 26% | 26% | 26% | 26% | 26% |
| | | Prune | 0 % | 0% | 0% | 25% | 25% |
| SVHN (VGG) | Clean Accuracy | Path connection ($t = 0.1$) | 96% | 94% | 93% | 91% | 90% |
| | | Fine-tune | 94% | 93% | 92% | 90% | 81% |
| | | Train from scratch | 88% | 82% | 76% | 68% | 28% |
| | | Noisy model ($t = 0$) | 33% | 33% | 33% | 33% | 33% |
| | | Noisy model ($t = 1$) | 37% | 37% | 37% | 37% | 37% |
| | | Prune | 94% | 93% | 92% | 90% | 89% |
| | Injection Accuracy | Path connection ($t = 0.1$) | 0% | 0% | 0% | 0% | 0% |
| | | Fine-tune | 0% | 0% | 25% | 0% | 25% |
| | | Train from scratch | 0% | 0% | 0% | 0% | 0% |
| | | Noisy model ($t = 0$) | 24% | 24% | 24% | 24% | 24% |
| | | Noisy model ($t = 1$) | 29% | 29% | 29% | 29% | 29% |
| | | Prune | 0% | 25% | 0% | 25% | 25% |

# G  EXTENSIONS OF PATH CONNECTION TO DIFFERENT SETTINGS

In Sections 3.2 and 3.3, we consider the scenario where the two given models are tampered using in the same way – using the same poisoned dataset for backdoor attack and the same targeted images for error-injection attack. Here we discuss how to apply path connection to the case when only one tampered model is given. In addition, we show that the the resilient effect of path connection against these attacks still hold when the two given models are tampered in a different way.

**Backdoor and error-injection attacks given one tampered model**  We propose to first fine-tune the model using the bonafide data and then connect the original model with the fine-tuned model. The fine-tuning process uses 2000 images with 100 epochs. The path connection results are shown in Figure A7. The start model is a backdoored model with high accuracy for triggered images. The end model is a fine-tuned model where the triggers do not have any effects to cause any misclassification. We can see that through path connection, we can eliminate the influence of the triggers quickly in some cases. For example, with 250 images, the error rate of triggered images reaches 100% at $t = 0.25$ while the clean accuracy at this point is lower than the fine-tuned model at $t = 1$.

Similarly, for the case of error-injection attack, we first fine-tune the model using the bonafide data and then connect the original model with the fine-tuned model. We follow the same settings of the backdoor attack for the finetuning and path connection. The performance of the one tampered model case is shown in Figure A8(a). We can see that the effects of injected errors can be eliminated quickly through path connection while the clean accuracy is kept high.

**Backdoor and error-injection attacks given two differently tampered models**  If the given two backdoored models are trained with different poisoned datasets (e.g. different number of poisoned images), the path connection method works as well in this case. We train two backdoored models by poisoning 30% and 10% of the training dataset, respectively. The performance of the path connection between the two models are shown in Figure A9. We can see that the connection can quickly remove the adversarial effects of backdoor attacks.

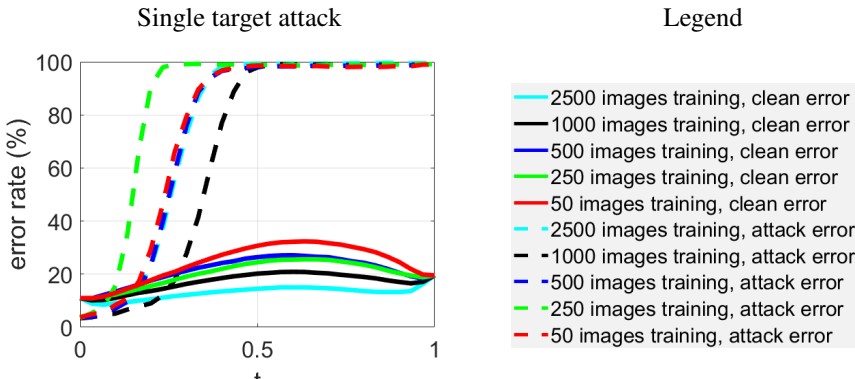

Figure A7: Error rate under backdoor attack on path connection for CIFAR-10 (VGG). The error rate of clean/triggered samples means the standard-test-error/attack-failure-rate, respectively.

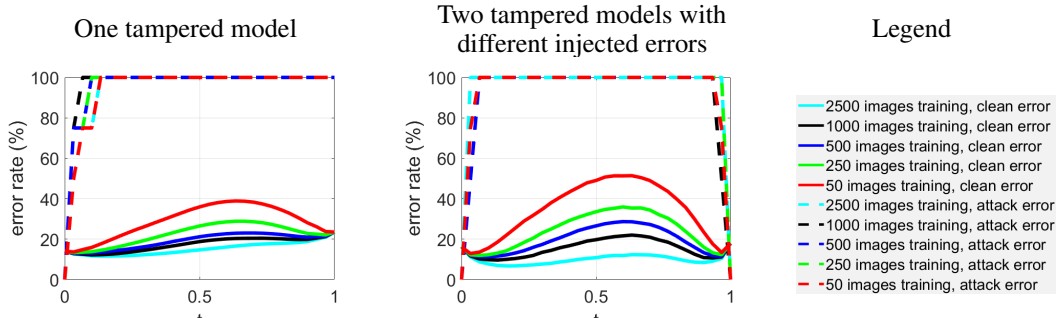

Figure A8: Error rate against error-injection attack on the connection path for CIFAR-10 (VGG). The error rate of clean/targeted samples means standard-test-error/attack-failure-rate, respectively.

If the two given models with injected errors are trained with different settings (e.g. different total number of training images to inject the errors), the path connection method works as well in this case. For the start and end models, the number of injected errors is set to 4. The number of images with the same classification requirement is set to 996 for the start model, and 1496 for the end model, respectively. The performance of path connection is shown in Figure A8(b). We can observe that it is able to obtain a robust model with high clean accuracy.

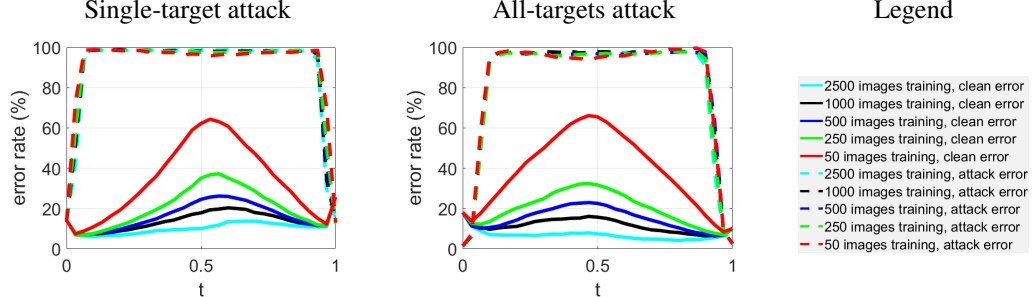

Figure A9: Error rate against backdoor attack on the connection path for CIFAR-10 (VGG). The error rate of clean/triggered samples means the standard-test-error/attack-failure-rate, respectively.

## H    MODEL WEIGHT SPACE EXPLORATION

To explore the model weight space, we add Gaussian noise to the weights of a backdoored model to generate 1000 noisy models of the backdoored model at $t = 0$. The standard normal Gaussian noise has zero mean and a standard deviation of the absolute difference between the two end models on the

path. The distribution of clean accuracy and backdoor accuracy of these noisy models are reported in Figure A10 (a). The results show that the noisy models are not ideal for attack mitigation and model repairing since they suffer from low clean accuracy and high attack success rate. In other words, it is unlikely, if not impossible, to find good models by chance. We highlight that finding a path robust to backdoor attack between two backdoored models is highly non-intuitive, considering the high failure rate of adding noise. We can observe similar phenomenon for the injection attack in Figure A10 (b).

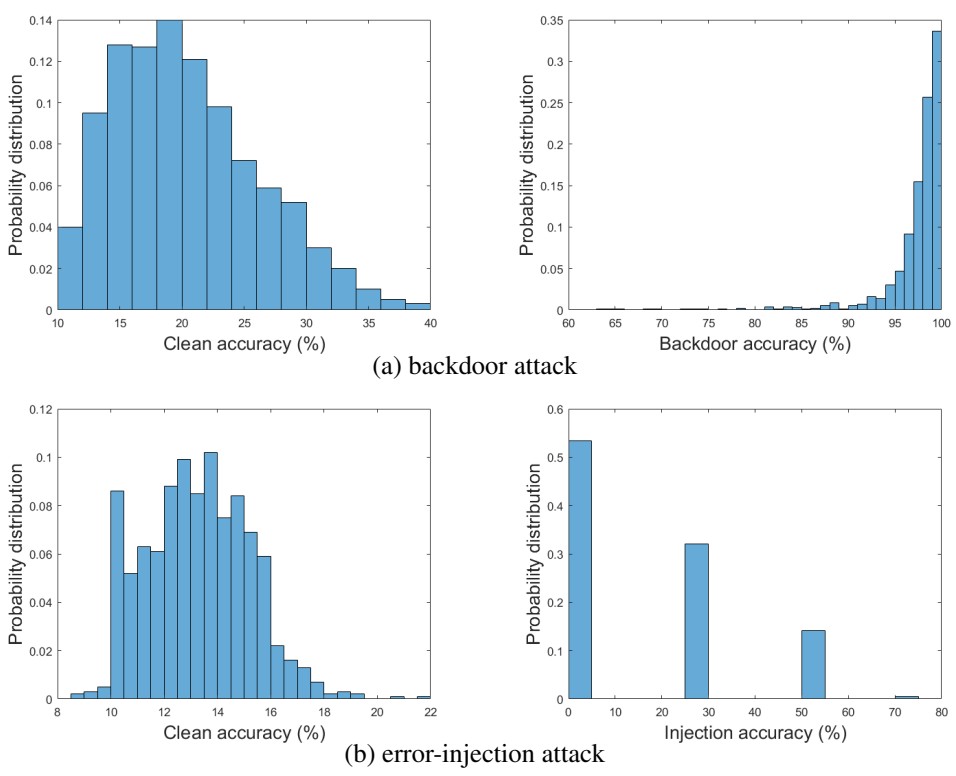

Figure A10: Clean and attack accuracy distribution for 1000 noisy models.

## I    DATA SIMILARITY OF INPUT GRADIENTS

To provide a technical explanation on why our path connection approach can mitigate backdoor and injection attacks, we compare the similarity of input gradients between the models on the connection path ($t \in (0, 1)$) and the end models ($t = 0$ or $t = 1$) in terms of clean data and tampered data. The rationale of inspecting input gradient similarity can be explained using the first-order approximation of the training loss function. Let $l(x|w_t)$ denote the training loss function of a data input $x$ given a model $w_t$ on the connection path, where $t \in [0, 1]$. Then the first-order Taylor series approximation on $l(x|w_t)$ with respect to a data sample $x_i$ is $l(x|w_t) = l(x_i|w_t) + \nabla_x l(x_i|w_t)^T (x - x_i)$, where $\nabla_x l(x_i|w_t)$ is the gradient of $l(x|w_t)$ when $x = x_i$. Based on the flat loss of mode connectivity, we can further assume $l(x_i|w_t)$ is a constant for any $t \in [0, 1]$. Therefore, for the same data sample $x_i$, the model $w_t$ ($t \in (0, 1)$) will behave similarly as the end model $w_0$ (or $w_1$) if its input gradient $\nabla_x l(x_i|w_t)$ is similar to $\nabla_x l(x_i|w_0)$ (or $\nabla_x l(x_i|w_1)$). Figure A11 shows the average cosine similarity distance of the input gradients between the models on the path and the end models for the backdoor attack and the injection attack. The pairwise similarity metric for each data sample is defined as $m = |s - 1|/2$, where $s$ is the cosine similarity of the input gradients between the models on the path and the end models. Smaller $m$ means higher similarity of the input gradients. Comparing the minimum of the solid and dashed curves of different colors on the path respectively, which corresponds to the similarity to either one of the two end models, we find that the similarity of clean data are consistently higher than that of tampered data. Therefore, the sanitized models on the

path can maintain similar clean data accuracy and simultaneously mitigate adversarial effects as these models are dissimilar to the end models for the tampered data.

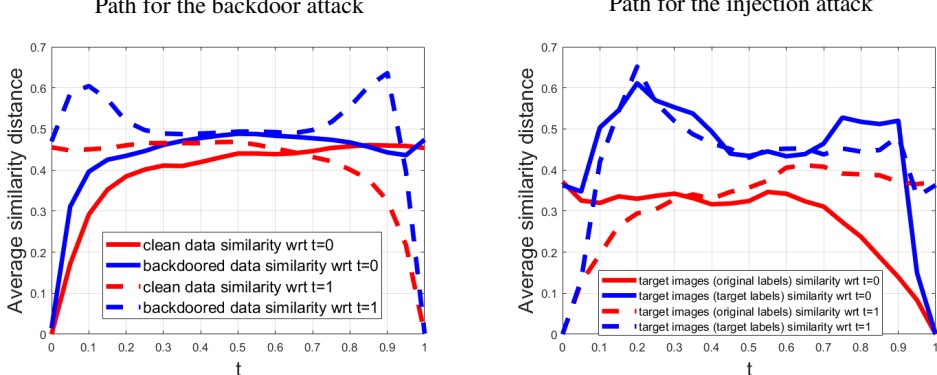

Figure A11: Similarity distance between the models on the path and the two end models for CIFAR-10 (VGG). Smaller distance value means higher similarity.

## J ADAPTIVE ATTACKS

We consider the advanced attack setting where the attacker knows path connection is used for defense but cannot compromise the bonafide data. Furthermore, we allow the attacker to use the *same* path training loss function as the defender.

**Backdoor attack.** To attempt breaking path connection, the attacker first separately trains two backdoored models with one poisoned dataset. Then the attacker uses the same poisoned dataset to connect the two models and hence compromises the models on the path. Note that when training this tampered path, in addition to learning the path parameter $\theta$, the start and end models are not fixed and they are fine-tuned by the poisoned dataset. Next the adversary releases the start and end models ($t = 0, 1$) from this tampered path. Finally, the defender trains a path from these two models with bonafide data. We conduct the advanced (path-aware) single-target backdoor attack experiments on CIFAR-10 (VGG) by poisoning 10% of images in the training set with a trigger. Figure A12 (a) shows the entire path has been successfully compromised due to the attacker's poisoned path training data, yielding less than 5% attack error rate on 10000 test samples. Figure A12 (b) shows the defense performance with different number of clean data to connect the two specific models released by the attacker. We find that path connection is still resilient to this advanced attack and most models on the path (e.g. $t \in [0.25, 0.75]$) can be repaired. Although this advanced attack indeed decreases the portion of robust models on the path, it still does not break our defense. In Table A4, we also compare the generalization and defense performance and demonstrated that path connection outperforms other approaches. Moreover, in the scenario where two tampered models are close in the parameter space (in the extreme case they are identical), we can leverage the proposed path connection method with one tampered model to alleviate the adversarial effects. The details are discussed in the "Extensions" paragraph and Appendix G.

Table A4: Performance against path-aware single-target backdoor attack on CIFAR-10 (VGG).

| Method / Bonafide data size | Clean accuracy | | | | | Backdoor accuracy | | | | |
|---|---|---|---|---|---|---|---|---|---|---|
| | 2500 | 1000 | 500 | 250 | 50 | 2500 | 1000 | 500 | 250 | 50 |
| Path connection ($t = 0.27$) | 90% | 87% | 83% | 81% | 75% | 3.8% | 2.9% | 3.6% | 4.2% | 5.6% |
| Fine-tune | 88% | 84% | 82% | 80% | 69% | 4.1% | 4.6% | 4.4% | 3.9% | 5.9% |
| Train from scratch | 58% | 48% | 36% | 28% | 20% | 0.6% | 0.5% | 0.9% | 1.7% | 1.9% |
| Noisy model ($t = 0$) | 38% | 38% | 38% | 38% | 38% | 91% | 91% | 91% | 91% | 91% |
| Noisy model ($t = 1$) | 35% | 35% | 35% | 35% | 35% | 86% | 86% | 86% | 86% | 86% |
| Prune | 87% | 85% | 83% | 81% | 79% | 29% | 48% | 69% | 77% | 81% |

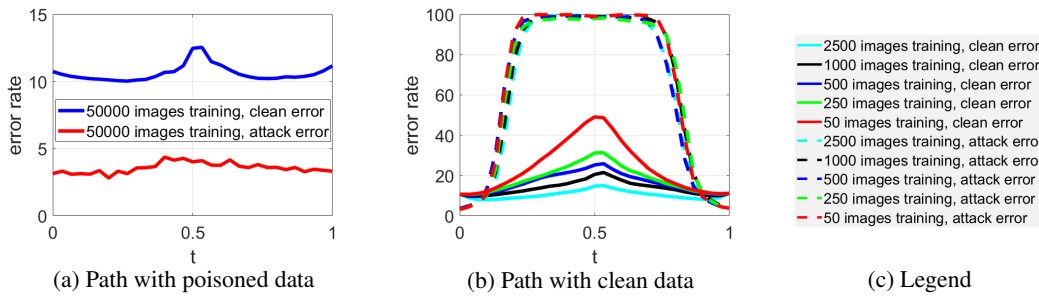

Figure A12: Error rate against path-aware single-target backdoor attacks for CIFAR-10 (VGG).

**Error-injection attack.** For the advanced attack in the error-injection setting, the attacker first trains two models with injected errors and then connects the two models with clean data. Then, the attacker tries to inject errors to the three models $w_1$, $w_2$ and $\theta$. Note that based on Equation (3) or (4), all the models on the path can be expressed as a combination of these three models. Thus, the whole path is expected to be tampered. Next, the attacker releases the start and end models from the "bad" path. Finally, the defender trains a path from these two models with bonafide data. We conduct the advanced (path-aware) error-injection attack experiments on CIFAR-10 (VGG) by injecting 4 errors. Figure A13 (a) shows that the entire path has been successfully compromised by the advanced path-aware injection. While the clean error rate is stable across the path, at least 3 out of 4 injected errors (corresponding to 25% attack error rate) yield successful attacks. After training a path to connect the two released models with the bonafide data, the clean error and attack error are shown in Figure A13 (b). We can observe that path connection can effectively eliminate the injected errors on the path (i.e., high attack error rate).

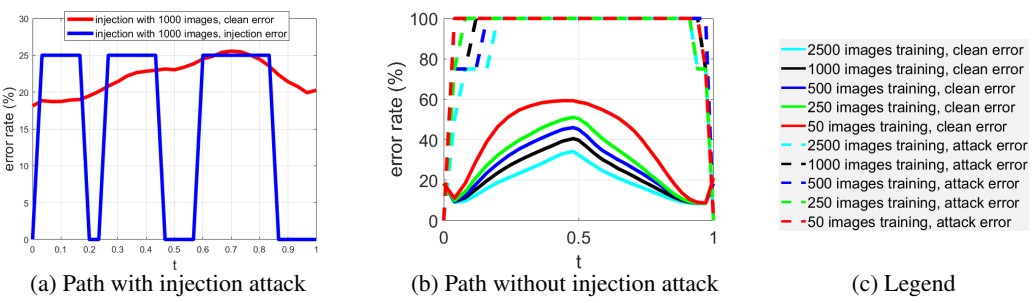

Figure A13: Error rate against path-aware error-injection attacks for CIFAR-10 (VGG).

## K  IMPLEMENTATION DETAILS FOR EVASION ATTACK AND ADVERSARIAL TRAINING

**Evasion attack**  After training the connection with the training set for 100 epochs, adversarial examples of the whole test set generated by the $\ell_\infty$-norm based PGD method (Madry et al., 2018) with attack strength $\epsilon = 8/255$ and 10 attack iterations are used to test the performance of the path connection.

**Adversarial training**  For adversarial training, at each model weight update stage, we first generate adversarial examples with the PGD method (Madry et al., 2018). The attack strength is set to $\epsilon = 8/255$ with 10 attack iterations. After the adversarial examples are obtained, we then update the model weights based on the training losses induced by these adversarial examples with their correct labels.

**Input Hessian**  As the adversarial examples perform small perturbations around the clean images to increase the robustness loss function, thus incurring a misclassification, it relates to the Hessian

matrix of the loss function with reference to the input images. A Hessian matrix (or Hessian) is the second-order partial derivatives of a scalar-valued function, describing the local curvature of a function with many variables. In general, large Hessian spectrum means the function reaches a sharp minima, thus leading to a more vulnerable model as the input can leave this minima with small distortions. By contrast, in the case of flat minima with small Hessian spectrum, it takes more efforts for the input to leave the minima.

As we find the robustness loss barrier on the path, we are interested in the evolution of the input Hessian for the models on the path. We uniformly pick the models on the connection and compute the largest eigenvalue of the Hessian w.r.t. to the inputs. To deal with the high dimension difficulties for the Hessian calculation, the power iteration method (Martens & Sutskever, 2012) is adopted to compute the largest eigenvalue of the input Hessian with the first-order derivatives obtained by back-propagating. Unless otherwise specified, we continue the power iterations until reaching a relative error of 1E-4.

For ease of visual comparison, in Figure 4, we plot the log value of the largest eigenvalue of the input Hessian together with the error rate and loss for clean images and adversarial examples. We note that the Pearson correlation coefficient (PCC) is indeed computed using the original largest eigenvalue of input Hessian and robustness loss. As demonstrated in Figure 4, the evolution of the input Hessian is very similar to the change of the loss of adversarial examples. As we can see, the largest eigenvalue on the path does not necessarily have a high correlation with the error rate of the clean images in the training set and test set or the adversarial examples. Instead, it seems to be highly correlated with the loss of adversarial examples as they share very similar shapes. This inspires us to explore the relationship between the largest eigenvalue of the input Hessian and the robustness loss.

## L    PROOF OF PROPOSITION 1

For simplicity, here we drop the the notation dependency on the input data sample $x$. It also suffices to consider two models $w := w(t)$ and $w + \Delta w := w(t + \Delta t)$ for some small $\Delta t$ on the path for our analysis. We begin by proving the following lemma.

**Lemma 1.** *Given assumption (a), for any vector norm $\| \cdot \|$, for any data sample $x$,*

$$\|\nabla_x l\left(w + \Delta w, x\right)\| - \|\nabla_x l\left(w, x\right)\| = 0 \tag{A1}$$

*Proof.* With assumption (a), we have $l\left(w + \Delta w, x\right) = l\left(w, x\right) = $ const. Differentiating at both sides with respect to $x$ and then applying the vector norm gives the lemma. $\square$

As the adversarial perturbation is generated using the PGD method (Madry et al., 2018) with an $\epsilon$-$\ell_\infty$ ball constraint, the optimal first-order solution of the perturbation is

$$\delta^* = \epsilon \cdot \frac{\nabla f\left(x\right)}{\|\nabla f\left(x\right)\|} \tag{A2}$$

Assume the PGD method can well approximate the robustness loss, the difference of the robustness losses of the two models $w$ and $w + \Delta w$ on the path can be represented as

$$\max_{\|\delta\| \leq \epsilon} l\left(w + \Delta w, x + \delta\right) - \max_{\|\delta\| \leq \epsilon} l\left(w, x + \delta\right)$$

$$= l\left(w + \Delta w, x + \delta^*_{w+\Delta w}\right) - l\left(w, x + \delta^*_w\right) \tag{A3}$$

$$= \left[l\left(w + \Delta w, x\right) + \epsilon \frac{\nabla_x l\left(w + \Delta w, x\right)^T \nabla_x l\left(w + \Delta w, x\right)}{\|\nabla_x l\left(w + \Delta w, x\right)\|} + \frac{\epsilon^2}{2} \frac{\nabla_x l\left(w + \Delta w, x\right)^T}{\|\nabla_x l\left(w + \Delta w, x\right)\|} H_{w+\Delta w} \frac{\nabla_x l\left(w + \Delta w, x\right)}{\|\nabla_x l\left(w + \Delta w, x\right)\|}\right]$$

$$- \left[l\left(w, x\right) + \epsilon \frac{\nabla_x l\left(w, x\right)^T \nabla_x l\left(w, x\right)}{\|\nabla_x l\left(w, x\right)\|} + \frac{\epsilon^2}{2} \frac{\nabla_x l\left(w, x\right)^T}{\|\nabla_x l\left(w, x\right)\|} H_w \frac{\nabla_x l\left(w, x\right)}{\|\nabla_x l\left(w, x\right)\|}\right] + O(\epsilon^3) \tag{A4}$$

$$\approx l\left(w + \Delta w, x\right) - l\left(w, x\right) + \epsilon \left[\|\nabla_x l\left(w + \Delta w, x\right)\| - \|\nabla_x l\left(w, x\right)\|\right]$$

$$+ \frac{\epsilon^2}{2} \left[\frac{\nabla_x l\left(w + \Delta w, x\right)^T}{\|\nabla_x l\left(w + \Delta w, x\right)\|} H_{w+\Delta w} \frac{\nabla_x l\left(w + \Delta w, x\right)}{\|\nabla_x l\left(w + \Delta w, x\right)\|} - \frac{\nabla_x l\left(w, x\right)^T}{\|\nabla_x l\left(w, x\right)\|} H_w \frac{\nabla_x l\left(w, x\right)}{\|\nabla_x l\left(w, x\right)\|}\right] \tag{A5}$$

$$= \frac{\epsilon^2}{2} \left[\frac{\nabla_x l\left(w + \Delta w, x\right)^T}{\|\nabla_x l\left(w + \Delta w, x\right)\|} H_{w+\Delta w} \frac{\nabla_x l\left(w + \Delta w, x\right)}{\|\nabla_x l\left(w + \Delta w, x\right)\|} - \frac{\nabla_x l\left(w, x\right)^T}{\|\nabla_x l\left(w, x\right)\|} H_w \frac{\nabla_x l\left(w, x\right)}{\|\nabla_x l\left(w, x\right)\|}\right] \tag{A6}$$

The first equality uses the definition of optimal first-order solution of the robustness loss. Using equation A2 and Taylor series expansion on $\ell(\cdot, \cdot)$ with respect to its second argument gives equation A4. Rearranging equation A4 and using Assumption (b) (i.e., ignoring the $O(\epsilon^3)$ term in equation A4) gives equation A5. Finally, Based on equation A5, the term $l(w + \Delta w, x) - l(w, x) = 0$ due to Assumption (a). Then, applying Lemma 1 to equation A5 gives equation A6.

Without loss of generality, the following analysis assumes $\| \cdot \|$ to be the Euclidean norm and consider the case when $\frac{\nabla_x l(w,x)^T v}{\|\nabla_x l(w,x)\|} = c$. The results can generalize to other norms since the correlation is invariant to scaling factors. With assumption (c), for any input Hessian $H$ and its largest eigenvector $v$, we have

$$
\begin{aligned}
&\frac{\nabla_x l(w,x)^T}{\|\nabla_x l(w,x)\|} H \frac{\nabla_x l(w,x)}{\|\nabla_x l(w,x)\|} \\
&= \left( \frac{\nabla_x l(w,x)}{\|\nabla_x l(w,x)\|} - v + v \right)^T H \left( \frac{\nabla_x l(w,x)}{\|\nabla_x l(w,x)\|} - v + v \right) \\
&= \left( \frac{\nabla_x l(w,x)}{\|\nabla_x l(w,x)\|} - v \right)^T H \left( \frac{\nabla_x l(w,x)}{\|\nabla_x l(w,x)\|} - v \right) + 2 \frac{\nabla_x l(w,x)^T}{\|\nabla_x l(w,x)\|} Hv - \lambda_{\max} \\
&= (2c - 1)\lambda_{\max} + h_x,
\end{aligned}
\tag{A7}
$$

where $h_x := \left( \frac{\nabla_x l(w,x)}{\|\nabla_x l(w,x)\|} - v \right)^T H \left( \frac{\nabla_x l(w,x)}{\|\nabla_x l(w,x)\|} - v \right)$ and the last equality uses the fact that $Hv = \lambda_{\max} \cdot v$ and assumption (c). Note that the lower bound is tight when $v$ and $\nabla_x l(w,x)$ are collinear, i.e., $v = \frac{\nabla_x l(w,x)^T}{\|\nabla_x l(w,x)\|}$, as in this case $h_x = 0$ and $c = 1$. On the other hand, using the definition of largest eigenvalue gives an upper bound

$$
\frac{\nabla_x l(w + \Delta w, x)^T}{\|\nabla_x l(w + \Delta w, x)\|} H \frac{\nabla_x l(w + \Delta w, x)}{\|\nabla_x l(w + \Delta w, x)\|} \leq \lambda_{\max}.
\tag{A8}
$$

Finally, applying equation A8 and equation A7 to equation A6, we have

$$
\max_{\|\delta\| \leq \epsilon} l(w + \Delta w, x + \delta) - \max_{\|\delta\| \leq \epsilon} l(w, x + \delta) \leq \frac{\epsilon^2}{2} [\lambda_{\max}(t + \Delta t) - (2c - 1)\lambda_{\max}(t) - h_x(t)]
\tag{A9}
$$

and

$$
\max_{\|\delta\| \leq \epsilon} l(w + \Delta w, x + \delta) - \max_{\|\delta\| \leq \epsilon} l(w, x + \delta) \geq \frac{\epsilon^2}{2} [(2c - 1)\lambda_{\max}(t + \Delta t) - \lambda_{\max}(t) + h_x(t + \Delta t)]
\tag{A10}
$$

As $v$ becomes more aligned with $\nabla_x l(w,x)$, $c$ increases toward 1 and $h_x$ decreases toward 0, implying $2c - 1 \approx 1$. Therefore, in this case we have $\max_{\|\delta\| \leq \epsilon} l(w + \Delta w, x + \delta) - \max_{\|\delta\| \leq \epsilon} l(w, x + \delta)$ as a linear function of $\lambda_{\max}(t + \Delta t) - \lambda_{\max}(t)$ and $\max_{\|\delta\| \leq \epsilon} l(w(t), x + \delta) \sim \lambda_{\max}(t)$.

If the higher order term $O(\epsilon^3)$ in equation A4 is non-negligible, which means Assumption (b) does not hold, then the following analysis will have an extra offset term of the order $O(\epsilon)$ as $c$ increases toward 1, i.e., $\max_{\|\delta\| \leq \epsilon} l(w(t), x + \delta) \sim \lambda_{\max}(t) + O(\epsilon)$.

# M  ROBUST CONNECTION AND MODEL ENSEMBLING

**Robust connection**  To train a robust path, we minimize the expected maximum loss on the path like the following,

$$
\min_\theta E_{t \sim U(0,1)} \left[ \max_{\delta \in S} L(\phi_\theta(t), \mathbf{x} + \delta) \right]
\tag{A11}
$$

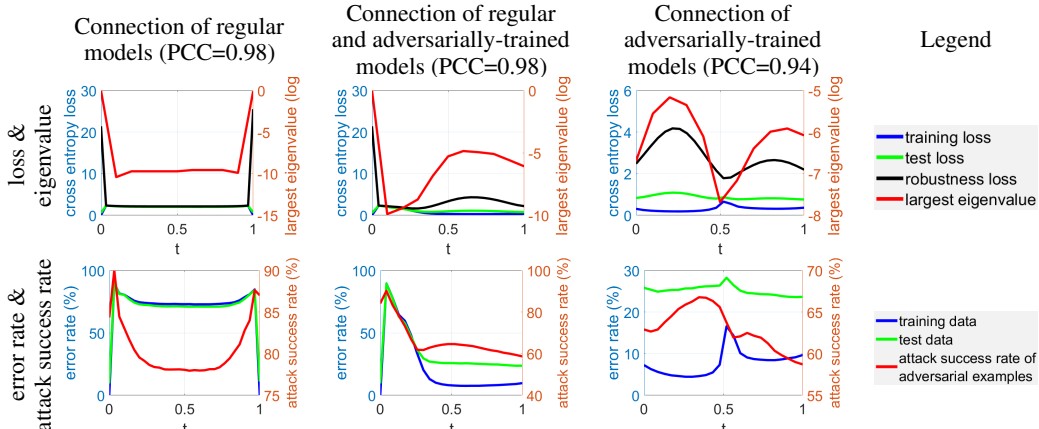

Figure A14: Loss, error rate, attack success rate and largest eigenvalue of input Hessian on the path connecting different model pairs on CIFAR-10 (VGG) using robust connection. The path is obtained with robust training method. The error rate of training/test data means standard training/test error, respectively. There is still a robustness loss barrier between non-robust and robust models, but not there is no robustness loss barrier between robust model pair and non-robust model pair (verified by small loss variance and flat attack success rate on the path). There is also a high correlation between the robustness loss and the largest eigenvalue of input Hessian, and their Pearson correlation coefficient (PCC) is given in the title of each plot.

where

$$S = \left\{ \delta \,\middle|\, \mathbf{x} + \delta \in [0,1]^d, \|\delta\|_\infty \le \epsilon \right\} \tag{A12}$$

To solve the problem, we first sample $\tilde{t}$ from the uniform distribution $U(0,1)$ and we can obtain the model $\phi_\theta(\tilde{t})$. Based on the model $\phi_\theta(\tilde{t})$, we find the perturbation maximizing the loss within the range $S$,

$$\max_{\delta \in S} L(\phi_\theta(\tilde{t}), \mathbf{x} + \delta) \tag{A13}$$

We can use projected gradient descent method for the maximization.

$$\delta_{k+1} = \prod_S \left( \delta_k + \eta \cdot \mathrm{sgn}\left( \nabla_\delta L(\phi_\theta(\tilde{t}), \mathbf{x} + \delta) \right) \right), \tag{A14}$$

where $\prod_S$ denotes the projection to the feasible perturbation space $S$, and $\mathrm{sgn}(\cdot)$ denotes element-wise sign function taking the value of either $1$ or $-1$.

After finding the perturbation $\hat{\delta}$ maximizing the loss, we would minimize the expectation. At each iteration, we make a gradient step for $\theta$ as follows,

$$\theta = \theta - \eta \cdot \nabla_\theta L(\phi_\theta(\tilde{t}), \mathbf{x} + \hat{\delta}) \tag{A15}$$

We show the robust connection for a pair of non-robust and non-robust models, a pair of non-robust and robust models, and a pair of robust models in Figure A14. We can observe that with the robust training method, there is still a robustness loss barrier between the non-robust and robust models. However, for the robust connection of robust model pair and non-robust (regular) model pairs, there is no robustness loss barrier, verified by small loss variance and flat attack success rate on the path. Our results also suggest that there is always a loss barrier between non-robust and robust models, no matter using standard or robust loss functions for path connection, as intuitively the two models are indeed not connected in their respective loss landscapes. Moreover, the attack success rate of adversarial examples are relatively small on the whole path compared with the robust connection of non-robust and robust models.

**Model ensembling**  Here we test the performance of (naive) model ensembling against evasion attacks. Given two untampered and independently trained CIFAR-10 (VGG) models, we first build a regular connection of them. Then we randomly choose models on the path (randomly choose the

value of $t$) and take the average output of these models as the final output if given an input image for classification. The adversarial examples are generated based on the start model ($t = 0$) or end model ($t = 1$) and we assume the attacker does not have any knowledge about the connection path nor the models on the path. We use these adversarial examples to test the performance of the model ensembling with the models on the connection. The attack success rate of adversarial examples can decrease from 85% to 79%. The defense improvement of this naive model ensembling strategy is not very significant, possibly due to the well-known transferrability of adversarial examples (Papernot et al., 2016; Su et al., 2018). Similar findings can be concluded for the robust connection method.

## N  EXPERIMENTAL RESULTS ON CIFAR-100

To evaluate the proposed path connection method on more complicated image recognition benchmarks, we demonstrate its performance against backdoor and error-injection attacks on CIFAR-100 as shown in Figure A15. The experimental setting is similar to that of CIFAR-10, where the two end models are backdoored/error-injected, and the connection is trained with various bonafide data sizes. We can observe that our method is still able to remove the adversarial effects of backdooring or error-injection and repair the model.

For the evasion attack, we investigate two cases, 1) the connection of two regular models and 2) the connection of regular and adversarially-trained models. The performance of adversarial training on CIFAR-100 is not as significant as that on CIFAR-10. So we do not investigate the connection of two adversarially-trained models. Their loss and eigenvalue on the path are shown in Figure A16. We can observe that there is also a high correlation between the robustness loss (loss of adversarial examples) and the largest eigenvalue of input Hessian.

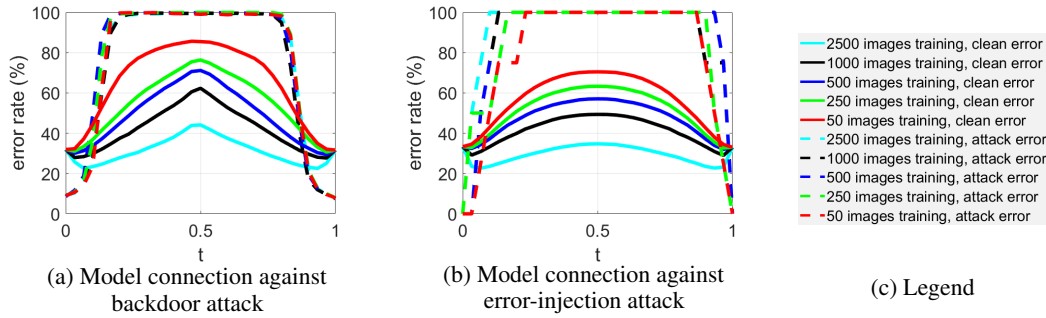

(a) Model connection against backdoor attack

(b) Model connection against error-injection attack

(c) Legend

Figure A15: Error rate against poison and error-injection attacks for CIFAR-100 (VGG).

## O  FINE-TUNING WITH VARIOUS HYPER-PARAMETERS

We demonstrate the performance of fine-tuning with various hyper-parameter configurations in this section. For CIFAR-10 (VGG), we perform fine-tuning with different learning rate and the number of total epochs with the bonafide data of 2500 images and 1000 images, respectively. The clean accuracy and attack accuracy are shown in Figure A17. We also plot the clean and attack accuracy obtained through our path connection method from Table 2 in Figure A17 as a reference.

As observed from Figure A17 (a), larger learning rate (such as 0.05) can decrease the attack accuracy more rapidly, but the clean accuracy may suffer from a relatively large degradation. Small learning rate (such as 0.01) can achieve high clean accuracy, but the attack accuracy may decrease with a much slower speed, leading to high attack accuracy when fine-tuning stops. This is more obvious if we use less bonafide data (reducing data size from 2500 to 1000) as shown in Figure A17 (b). Fine-tuning performs worse with lower clean accuracy and higher attack accuracy. Since fine-tuning is quite sensitive to these hyper-parameters, we conclude that it is not easy to choose an appropriate learning rate and the number of fine-tuning epochs, especially considering the user is not able to observe the attack accuracy in practice. On the other hand, in Figure A17 (a) our path connection method can achieve the highest clean accuracy. In Figure A17 (b), although the clean accuracy of $lr = 0.01$ is

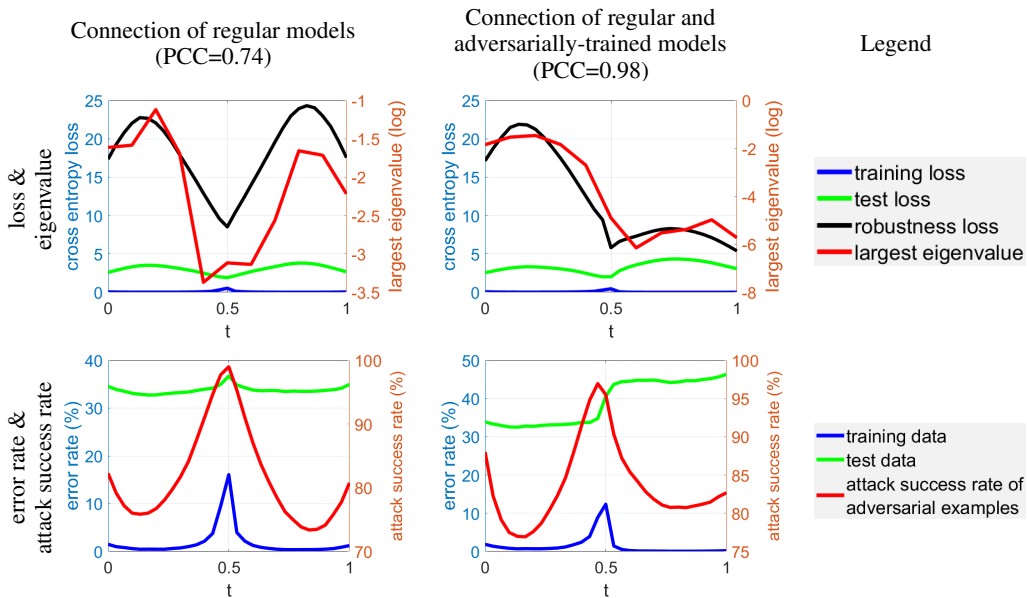

Figure A16: Loss, error rate, attack success rate and largest eigenvalue of input Hessian on the path connecting different model pairs on CIFAR-100 (VGG) using standard loss. The error rate of training/test data means standard training/test error, respectively. In all cases, there is no standard loss barrier but a robustness loss barrier. There is also a high correlation between the robustness loss and the largest eigenvalue of input Hessian, and their Pearson correlation coefficient (PCC) is reported in the title.

higher than that of path connection, its attack accuracy remains high (about 40%), which is much larger than that of path connection (close to 0%).

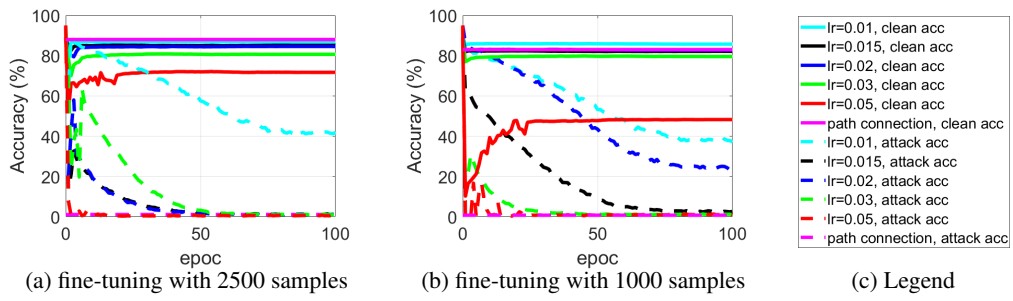

Figure A17: Test accuracy and attack accuracy for fine-tuning on CIFAR-10 (VGG).

## P    STABILITY ANALYSIS

In Appendix E, we perform multiple runs for the Gaussian noise experiment and only report the average accuracy. The variance is not reported since the average accuracy is able to demonstrate that the Gaussian noise method is not a good choice for removing adversarial effects.

To investigate the stability of the path connection method with respect to every possible factor, it will cost considerable amount of time and resource to run all experiment setups considering the various attack methods, datasets, and model architectures. So here we mainly perform one representative experiment setup with multiple runs and show their mean and standard deviation.

Figure A18 shows the error bars of the error rate computed over 10 runs for path connection against backdoor attack. The dataset is CIFAR-10 and the model architecture is ResNet. For each bonafide

data size, we train 10 connections with different hyper-parameter settings, that is, starting from random initializations and using various learning rates (randomly set learning rate to 0.005, 0.01 or 0.02). Their average value and standard deviation are shown in Table A5. We can observe that although the connection may start from different initializations and trained with different learning rates, their performance on the path are close with a relatively small variance, demonstrating the stability of our proposed method.

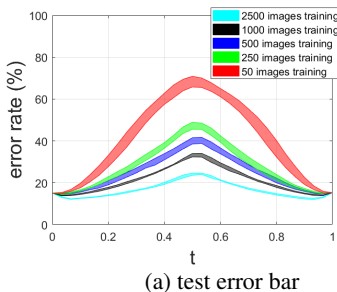
(a) test error bar

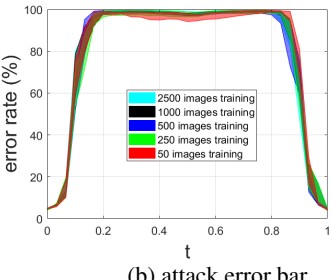
(b) attack error bar

Figure A18: Error rate against backdoor attack for CIFAR-10 (ResNet).

Table A5: Performance comparison of path connection against single-target backdoor attack. The clean/backdoor accuracy means standard-test-accuracy/attack-success-rate, respectively.

|  |  | data size | 2500 | 1000 | 500 | 250 | 50 |
|---|---|---|---|---|---|---|---|
| CIFAR-10 (ResNet) | Clean Accuracy | $t = 0.2$ | 87±0.21 % | 83 ±0.2 % | 80.4 ±0.72% | 78.4 ±0.82% | 69 ±1.9% |
|  |  | $t = 0.4$ | 80.5±0.37% | 73.5±0.19% | 67.7±1.0% | 62.1±1.0% | 40.5±2.5 % |
|  |  | $t = 0.6$ | 78.8±0.16% | 71.6±0.7% | 64.8±1.3 % | 58.8±1.4% | 36.9±1.4 % |
|  |  | $t = 0.8$ | 85.4 ±0.25% | 81.8±0.35% | 78.8±0.84 % | 76.8±1.1% | 64±2.9% |
|  | Backdoor Accuracy | $t = 0.2$ | 1.2±0.47 % | 1.6±0.54% | 1.3±0.64 % | 1.9±1% | 2.6±0.54 % |
|  |  | $t = 0.4$ | 0.8±0.18 % | 1.3±0.32% | 1.6±0.54% | 1.8±0.9% | 3.2 ±1.4% |
|  |  | $t = 0.6$ | 1.0±0.16% | 1.5±0.41% | 1.6±0.42% | 1.6±0.57% | 3.2±1.3% |
|  |  | $t = 0.8$ | 0.8±0.37% | 0.86±0.33% | 1.0±0.78 % | 1.1±0.52% | 1.8±0.3% |

## Q  STRATEGY TO CHOOSE THE PARAMETER $t$

In our proposed method, we need to choose a model on the path as the repaired model, that is, choosing the value of $t$. For different datasets/models, the best choice of $t$ may vary. So we discuss some general principles to choose an appropriate $t$ in this section.

We note that in practice the user is not able to observe the attack error rate as the user does not have the knowledge about certain attacks. If the user is able to observe the accuracy on the whole clean test set, we suggest the user to choose the model (a value of $t \in [0, 1]$) with a test accuracy $a - \Delta a$, where $a$ is the accuracy of the end model and $\Delta a$ represents a threshold. Based on the performance evaluations (Figures 2, 3, A4, A5, and A6), setting $\Delta a$ to 6% should be an appropriate choice, which is able to eliminate the effects of all attacks without significantly sacrificing the clean accuracy.

If the user is not able to access the accuracy of the whole clean test set, the user has to choose $t$ only based on the bonafide data. In this case, we suggest the user to use the $k$-fold cross-validation method to assess the test accuracy. This method first shuffles the bonafide data randomly and splits it into $k$ groups. Then one group is kept to test the accuracy on the learned path and the remaining $k - 1$ groups are used to train the path connection. The process is repeated for each group. We perform additional experiments with the 5-fold cross validation method for CIFAR-10 (VGG) and SVHN (ResNet). The average validation error rate on the hold-out set and attack error rate against backdoor attack is shown in Figure A19. The error-injection attack is easier to counter and hence we do not explore the error-injection experiments.

We can observe that since the validation data size reduces to a much smaller value (one fifth of the bonafide data size), the test error rate becomes less stable. But it generally follows the trends of the test error rate on the whole clean test set (see Figure A4). So by utilizing the $k$-fold cross-validation

method, we can obtain the test accuracy on the limited validation set. Then the aforementioned threshold method can be used to choose the model on the path. Here we suggest setting $\Delta a$ to 10%, a more conservative threshold than the former case, as the user now does not has access to the accuracy of the whole test set. We also note that the performance of bonafide data with 50 images has a large deviation from other data size settings, so we suggest to use a larger bonafide data size with the k-fold cross validation method.

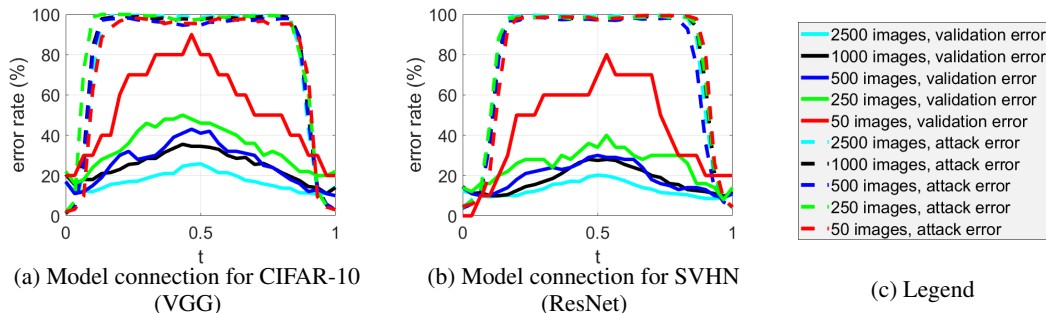

(a) Model connection for CIFAR-10 (VGG)

(b) Model connection for SVHN (ResNet)

(c) Legend

Figure A19: Average 5-fold cross validation error rate and attack error rate against backdoor attack.

