# OpenReview forum: "Bridging Mode Connectivity in Loss Landscapes and Adversarial Robustness"
_ICLR.cc/2020/Conference — Accept (Poster)_

### Official Review · AnonReviewer2 · 2019-10-22
**Official Blind Review #2**

**Rating:** 3

**Review:**

This paper proposes an adversarial defense method based on mode connectivity. The goal of the method is to repair tampered networks using a limited number of clean data examples. The authors consider two types of adversarial attacks: backdoor attacks and error-injection attacks. The proposed method takes two potentially tampered networks, then constructs a low-loss path connecting the weight vectors of the given models in the space of network parameters (the path is constructed using the small set of clean data examples), finally an intermediate point on the path is used as a weight vector corresponding to the “repaired” model. The authors analyze the properties of the paths and show that intermediate points on the mode-connecting paths deliver both high clean-data accuracy and low attack success rate. In the experiments the proposed method shows better results compared to baseline defense techniques including fine-tuning, training from scratch, and pruning followed by fine-tuning. The paper also analyzes evasion adversarial attacks from the perspective of mode-connectivity and observes the existence of barriers in the landscape of robustness loss on the paths connecting regular and adversarially-trained models.

I would like to note that the text of the paper is well-structured and clear. Another strength of paper is that the authors consider multiple supporting experimental settings and extensions of the proposed method. The considered cases include adaptive attacks (the settings in which the attacker is aware about the employed defense method).

Although there are many strong sides of this paper, I have identified several serious flaws in the justification of the approach and the results. Thus, I consider the paper to be marginally below the acceptance threshold. I am willing to increase the score if the authors address my concerns expressed below.

1) What is the conceptual difference between using mode-connectivity and fine-tuning?  Both methods try to find a point in the weight space which (1) is close to the initial point and (2) delivers low loss on the bonafide data. What feature of the proposed method can explain its success? The “technical explanations” for the effectiveness of the proposed path connection method only show that the random sampling procedure fails to find models with high-clean-accuracy and low-attack-accuracy. However, this doesn’t necessarily mean that fine-tuning (which performs directed search) cannot find models which perform similarly to the models found by the proposed method (potentially the hyperparameters of the fine-tuning procedure can be tuned better). Given the aforementioned similarity of the proposed approach and fine-tuning, the explanation (either theoretical or empirical) for the effectiveness of the proposed method has to be provided in order to motivate the methodological proposal of the paper.

2) Please report the error bars of the accuracy computed over multiple runs of the experiments (according to Appendix E multiple runs were performed, but only the mean values of the accuracy are reported). Including the error bars would help to quantify the statistical significance of the results.

3) What strategy was used to choose the parameter t? The parameter has different values depending on the model/dataset. How should this value be chosen in general?

4) In the formulation of Proposition 1 it is first stated (assumption (c)) that the directions of the gradient w.r.t. x and the vector v are aligned (the normalized dot product of the vectors is >= c), however then the proposition requires that the vectors must be similar. Why assumption (c) has to be included in this formulation, if the proposition also requires a stronger condition? Either the assumptions or the claim of the proposition has to be formulated clearly. For example, it can be explicitly stated that the proportionality holds approximately (up to terms which go to 0 as c goes to 1). I also recommend to include a term which accounts for the tail of the Taylor expansion in equation A4 instead of using the “approximately equals” sign. Moreover, I recommend to include the discussion of the effect of this tail term on the proof and the statement of the preposition.

Additional minor comments, which do not affect the assessment:

In Section 3.1, paragraph 3 the description of Figure 1 reads as follows: “our path connection is trained using different portion of test data”. Another sentence below in the paragraph: “For example, path connection using merely 1000/2500 CIFAR-10 samples from the test set only reduces the test accuracy of VGG16 models by at most 10%/5%”. Which part of the dataset was used for finding the path connections? How the “test data”, which is claimed to be used for finding paths, is different from the test set used to evaluate the accuracy of the models?


**Experience Assessment:**

I have published one or two papers in this area.

**Review Assessment: Checking Correctness Of Derivations And Theory:**

I carefully checked the derivations and theory.

**Review Assessment: Checking Correctness Of Experiments:**

I carefully checked the experiments.

**Review Assessment: Thoroughness In Paper Reading:**

I read the paper thoroughly.

---

> ### Author Response · Authors · 2019-11-13
> **Response to Reviewer #2**
>
> We thank the reviewer for the valuable review comments and suggestions. Please find our point-by-point response to your concerns as follows.
>
> 1) Our proposed approach can indeed be understood as a type of fine-tuning method with the additional knowledge that there exists a flat-loss, high-accuracy path connecting two models, which can be characterized by simple parametric functions such as the Bezier curve. Therefore, given a limited number of bonafide data for model repairing, our results suggest that the prior knowledge of the existence of a "good path" helps finding a better model than native fine-tuning methods. We have included the discussion in the "Technical Explanations" paragraph of Section 3.2 in the revised version. Moreover, following your comment that "potentially the hyperparameters of the fine-tuning procedure can be tuned better", we have performed additional experiments with various hyperparameter configurations of the fine-tuning procedure. The results are reported in Appendix O of the revised version. We find that fine-tuning is quite sensitive to hyper-parameter changes. Moreover, our method still outperforms the best fine-tuning configurations in terms of higher test accuracy and lower attack accuracy.
>
> 2) We would like to clarify that the multiple runs in Appendix E only apply to the Gaussian noise setting. In addition, motivated by the reviewer's comment, we also perform a stability analysis of our path connection method with multiple runs in Appendix P of the revised version, implying that our results are indeed statistically significant.
>
> 3) Depending on whether the user can access to the accuracy of the whole test dataset or not, we have provided some principles for choosing the parameter t in Appendix Q of the revised version. Specifically, for the case that only bonafide data are available for choosing t, we suggest using k-fold cross validation principle on the bonafide data to select a proper t by setting a threshold on the validation accuracy. In Appendix Q, we have included the results of 5-fold cross validation of our path connection method.
>
> 4) We agree with the reviewer that assumption (c) is better suited as a statement rather than an "assumption". We have revised Proposition 1 accordingly. Furthermore, following your suggestion, we have included the analysis regarding the tail of the Taylor expansion in equation A4 of the revised version. We have also included the discussion of the effect of this tail term on the proof and the statement in the last paragraph of Appendix L of the revised version.
>
> 5) On "Test data": We are sorry that our description regarding the "test data" may cause some confusion, which has been addressed in the revised version. In our experiments we split the original test set into two parts: one for training the path (aka the bonafide data) and the other one for testing the generalization performance and adversarial robustness (aka the final test data with 5000 samples).
>
> We hope our responses addressed the reviewer’s concerns. We also would like to make the most of the openreview platform and are happy to take any additional questions the reviewer may have during the author rebuttal phase.

---

### Official Review · AnonReviewer1 · 2019-10-23
**Official Blind Review #1**

**Rating:** 8

**Review:**

This paper studies leveraging mode connectivity to defend against different types of attacks, including backdoor attacks, adversarial examples, and error-injection attacks. They perform a comprehensive evaluation to show the benign test accuracy and attack success rate over the models in the connected path between pairs of models with the same or different properties, e.g., both are attacked, both are benign, one attacked and one benign, etc., where the connected path is learned using existing algorithms to find the high-accuracy path over two different models. Their evaluation suggests that in certain attack scenarios, exploring the mode connectivity could help find a model that has a high benign accuracy, while with a significantly lower attack success rate than the models at the end points.

In general I like this paper. Although mode connectivity has been studied in the literature, to my knowledge, this is the first work to extensively study this topic in the context of attacks. An interesting part of the paper is their evaluation on backdoor attacks, where they show that with a very small number of training samples for fine-tuning, they are able to find a model with decent test accuracy, while the watermarks are removed. Meanwhile, the attacks studied in this work include various settings, i.e., poisoning attacks, error-injection attacks, evasion attacks, and also adaptive attacks where the adversary knows that the defender will use path connection to improve the robustness. These make this paper a good reference as a systematic study of their proposed topic.

However, my main question is that while the algorithm is pretty effective to defend against backdoor attacks and error-injection attacks, the results of evasion attacks are somehow negative. While it is helpful to show negative results if this is indeed the case, do you have some possible explanation why the models on the path are less robust than the two end models?

Another concern is that while the benign test accuracy of the model found by their algorithms looks good, both CIFAR-10 and SVHN evaluated in this paper have a small label set and may not be challenging enough. It would be great if the authors can provide some results on a more complicated image recognition benchmark, e.g., some intermediate-level datasets studied in previous work on attacks such as CIFAR-100.

-------------
Post-rebuttal comments

I thank the authors for clarifying my questions, and I keep my original score.
------------

**Experience Assessment:**

I have published in this field for several years.

**Review Assessment: Checking Correctness Of Derivations And Theory:**

I carefully checked the derivations and theory.

**Review Assessment: Checking Correctness Of Experiments:**

I carefully checked the experiments.

**Review Assessment: Thoroughness In Paper Reading:**

I read the paper thoroughly.

---

> ### Author Response · Authors · 2019-11-13
> **Response to Reviewer #1**
>
> We thank the reviewer for the valuable review comments and suggestions.
>
> The finding in Figure 4 that the models on the path are less robust than the two end models against evasion attack can be explained by the fact that we use standard loss to train the path connection,  which may find models that are less robust if the robust loss is not considered for path training. On the other hand, as discussed in Appendix M and shown in Figure A14, if we use robust loss for path training (which we call the robust connection method), we may be able to find more robust models on the path. We expect that as the field of adversarial robust training advances, future research toward developing provably effective loss functions can be useful for path training to find more robust models as well.
>
> Following your suggestion of adding CIFAR-100 dataset for performance evaluation, we have included the results in Appendix N of the revised version. The findings and conclusions are consistent with those on CIFAR-10 and SVHN.
>
> We hope our responses addressed the reviewer’s concerns. We also would like to make the most of the openreview platform and are happy to take any additional questions the reviewer may have during the author rebuttal phase.

---

### Official Review · AnonReviewer3 · 2019-10-24
**Official Blind Review #3**

**Rating:** 6

**Review:**

The paper repurposes results on mode connectivity (that is, minimal loss paths between local optima) to improve adversarial robustness. The idea is new (to me) and appears both elegant and effective. The approach provides a fast way to repair models that have been attacked.

More broadly, mode connectivity provides a powerful window into adversarial robustness and model choice. It constructs approximately optimal paths between models, where the path depends on a choice of loss. It thus allows to investigate how various models relate to each other from the perspective of various losses. The paper barely scratches the surface of this approach.

Comments:
The claim that all models on paths have similar test losses (bottom p8) seems a bit of a stretch given the bottom panels (middle and right) of Figure 4.
The paper is difficult to read; it somehow feels quite disjointed.
Proposition 1 is difficult to parse.


**Experience Assessment:**

I have read many papers in this area.

**Review Assessment: Checking Correctness Of Derivations And Theory:**

I assessed the sensibility of the derivations and theory.

**Review Assessment: Checking Correctness Of Experiments:**

I assessed the sensibility of the experiments.

**Review Assessment: Thoroughness In Paper Reading:**

I read the paper at least twice and used my best judgement in assessing the paper.

---

> ### Author Response · Authors · 2019-11-13
> **Response to Reviewer #3**
>
> We thank the reviewer for the valuable review comments and suggestions. We are sorry to learn that the reviewer feels our paper is difficult to read. We have improved the presentation in the revised version. As our paper proposed using the same model connectivity based method to investigate the adversarial robustness of three different DNN attack scenarios (backdoor, error-injection and evasion attacks), its scope is indeed broader than papers focusing on only one attack scenario.
>
> In Figure 4, we would like to note that the training/test loss is shown in the top row of the three subfigures, while the bottom row of the subfigures shows the corresponding error rate.  As the objective of path connection is minimizing the loss metric, one can observe that the training/test loss on the learned path are indeed quite flat, whereas their error rate are not necessarily as flat (although we do see that the loss and error rate are highly correlated).
>
> Following your comment that "Proposition 1 is hard to parse" and Reviewer #2's suggestion, we have also revised the presentation of Proposition 1.
>
> We hope our responses addressed the reviewer’s concerns. We also would like to make the most of the openreview platform and are happy to take any additional questions the reviewer may have during the author rebuttal phase.

---

### Decision · Program_Chairs · 2019-12-19

**Decision:**

Accept (Poster)

**Comment:**

This paper investigates improving robustness to adversarial examples by using mode connectivity in the loss function. The paper received three reviews by experts working in related areas. In a strongly positive review, R1 recommends Accept, but gives some specific technical questions. The authors submitted a response to these questions; in post-review comments, R1 was satisfied and maintained the highly positive review. R2 recommended Weak Reject and also asked specific technical questions, including some additional details on experiments, statistical significance, etc. The author response also convincingly responded to these concerns. R3 recommended Weak Accept but suggested improving the writing, which authors have done in their revision. Given that R1 and R3 are highly positive and R2's concerns were addressed in the response and revision, we now recommend (weak) Accept.